# Machine learning-informed and synthetic biology-enabled semi-continuous algal cultivation to unleash renewable fuel productivity

Bin Long [1], Bart Fischer[2], Yining Zeng[3], Zoe Amerigian [1], Qiang Li[1], Henry Bryant[2], Man Li[1,4], Susie Y. Dai[1,4] & Joshua S. Yuan [1,4 ✉]

Algal biofuel is regarded as one of the ultimate solutions for renewable energy, but its commercialization is hindered by growth limitations caused by mutual shading and high harvest costs. We overcome these challenges by advancing machine learning to inform the design of a semi-continuous algal cultivation (SAC) to sustain optimal cell growth and minimize mutual shading. An aggregation-based sedimentation (ABS) strategy is then designed to achieve low-cost biomass harvesting and economical SAC. The ABS is achieved by engineering a fast-growing strain, *Synechococcus elongatus* UTEX 2973, to produce limonene, which increases cyanobacterial cell surface hydrophobicity and enables efficient cell aggregation and sedimentation. SAC unleashes cyanobacterial growth potential with 0.1 g/L/ hour biomass productivity and 0.2 mg/L/hour limonene productivity over a sustained period in photobioreactors. Scaling-up the SAC with an outdoor pond system achieves a biomass yield of 43.3 g/m$^2$/day, bringing the minimum biomass selling price down to approximately $281 per ton.

[1] Department of Plant Pathology and Microbiology, Texas A&M University, College Station, TX 77843, USA. [2] Department of Agricultural Economics, Texas A&M University, College Station, TX 77843, USA. [3] Renewable Resources and Enabling Sciences Center, National Renewable Energy Laboratory, Golden, CO 80401, USA. [4] Synthetic and Systems Biology Innovation Hub (SSBiH), Texas A&M University, College Station, TX 77843, USA. ✉email: syuan@tamu.edu

Algae-based bioproduction represents one of the most energy- and carbon-efficient solutions for renewable fuels and $CO_2$ capture and utilization[1]. Despite significant potential and extensive efforts, the commercialization of algal biofuel has been hindered by limited sunlight penetration, poor cultivation dynamics, relatively low yield, and the absence of cost-effective industrial harvest methods[2–6]. Growth limitation caused by mutual shading and high dewatering costs are the major causes for these technical barriers[7–9]. Overcoming these challenges could enable viable algal biofuels to reduce carbon emissions, mitigate climate change, alleviate petroleum dependency, and transform the bioeconomy.

Algal antennae are highly efficient at absorbing almost all photons that hit them, leading to mutual shading[10]. The lack of thorough, quantitative understanding of mutual shading hinders light management and hampers algal growth potential. Precise light distribution pattern (LDP) prediction could guide an innovative cultivation design to unleash growth potential. However, most current computational models predict LDPs as one-dimensional light paths that are not representative of real-world LDPs[9–14]. Moreover, these models perform poorly at high cell concentrations with more severe light scattering and diffusive reflection[9–14]. Machine learning based on empirical training could overcome these challenges to achieve two- or even three-dimensional LDP predictions.

Besides the growth limitation, high costs and energy demands associated with harvesting and dewatering represent another significant technical barrier[3], creating an inherent dilemma between light availability and harvesting cost. High cell concentration is preferred for algal biomass harvesting to minimize cost per unit, but it will inevitably result in strong mutual shading that limits growth. Traditional methods like centrifugation, filtration, chemical flocculation, or bio-flocculation can make up as much as 30% of total costs and 50% of total energy use, which makes them impractical for frequent harvests to bypass mutual shading[3,5,15,16]. A cost-effective harvesting method is thus urgently needed to address this dilemma.

Here, we provide a solution for the aforementioned challenges with a cultivation design informed by machine learning and a synthetic biology-based platform implementation. First, we demonstrate machine learning as an effective LDP-prediction tool to assess light availability inside algal culture. Second, this light availability is used to predict cyanobacterial growth rates with a second machine learning model, GRM (growth rate prediction model). Together, the machine learning models allow accurate growth simulation and guide the design of a semi-continuous algal cultivation (SAC). SAC sustains optimal growth rates to minimize mutual shading and drastically increases biomass productivity. Third, and most importantly, we advance a strategy of aggregation-based sedimentation (ABS) for low-cost harvesting and cost-effective SAC implementation. The ABS is achieved by engineering *Synechococcus elongatus* UTEX 2973 (UTEX 2973) to produce limonene, which generates hydrophobic surface interaction and triggers cell aggregation for sedimentation. Moreover, the strain co-produces biomass as a potential fuel precursor and limonene as a value-added product. Scaling-up of the machine learning-informed SAC with an outdoor pond system also shows a high biomass productivity. The impacts of high yields from SAC and a simplified harvest method are assessed with a techno-economic analysis (TEA).

## Results

### Building machine learning models for LDP prediction.
Considering the asymmetry of light sources in most PBRs and raceway ponds, LDPs should be two-dimensional or even three-dimensional.

Here, we employed a two-dimensional grayscale image to represent the LDP, with grayscale values (GSV, range of 0 to 255 with 0 for black and 255 for white) representing light intensities (See details in Supplementary Method 1). The GSVs and light intensities showed a strong linear correlation with an average $R^2$ score of 0.969 across a wide range of cell concentrations, validating the approach (Fig. 1c). Next, we evaluated the effectiveness of machine learning in LDP prediction. The overall workflows of sample preparation and training processes are shown in Fig. 1a. Light intensity and cell concentration, the two major factors determining LDPs, were set as features and their corresponding LDPs were set as labels in training. We chose the support vector regression (SVR) algorithm to train due to its versatility[17–19], resulting in an LDP prediction model (LDPM, see details in Supplementary Method 2).

Evaluation of the LDPM prediction showed an $R^2$ score of 0.993 between all predicted LDPs and measured LDPs (Fig. 1d), indicating high prediction accuracy. A pixel-by-pixel evaluation of the entire LDP suggested that 94.4% of pixels achieved $R^2$ values > 0.90, and only 0.8% of pixels had $R^2$ values in the range of 0.79–0.85 (Fig. 1b and Supplementary Fig. 1), indicating precise predictions at most pixels. Pixels further away from the light source (row 12–row 18) showed relatively lower $R^2$ scores (Fig. 1b), presumably because of the increased complexity of the light pattern. Overall, the accurate LDP prediction proves the feasibility of using machine learning to model light availability inside algal cultures.

The high $R^2$ score (0.993) highlights the increased accuracy of the machine learning model over traditional mathematical models[10,13,14]. Furthermore, unlike mathematical models that can only predict one-dimensional light paths, machine learning-predicted LDPs can be two-dimensional or even three dimensional. Moreover, the upper cell concentration limit of the LDPM is about 3.9 g/L, which is higher than the limit of ~1 g/L presented in previous mathematical models[10,13,14]. The larger prediction range indicates that a machine learning-based strategy could address LDP prediction challenges caused by complex light scattering and interference at high cell concentrations. The methodology for LDP prediction proposed in this study could be transferred to any existing algal cultivation systems, such as indoor/outdoor PBRs or pond systems. The superior performance of the machine learning model–in particular, a larger prediction range and higher accuracy–enabled LDP outputs to be used to simulate growth curves using a second machine learning model. Such integration has not been achieved in previous studies and would guide cultivation optimization.

### LDP-enabled growth rate prediction.
The LDP prediction allowed us to quantify mutual shading and explore the impact of light availability on cyanobacterial growth. We found that the shading effect increased sharply when cells grew to a high concentration (Supplementary Fig. 2), similar to previous studies[13,20]. Cyanobacterial growth rates peaked when dark areas, defined as pixels with GSVs <25.5 (10% of the maximal value, see details in Supplementary Method 3), reached 43.1 ± 4.9% at all tested light conditions. The growth rate dropped drastically when dark areas reached a plateau ~65% (Supplementary Fig. 3). Specifically, when dark areas reached 43.1%, cell growth began to be inhibited by mutual shading. Such inhibition intensified after dark areas reached 65%. The strong correlation between light pattern and growth rates suggests that light availability is the primary factor determining cyanobacterial growth rates when nutrients are sufficient and temperature is controlled. The results are consistent with previous findings that light availability defines the growth potential for cyanobacteria given abundant nutrients[21,22]. More importantly, this quantitative understanding allowed us to develop

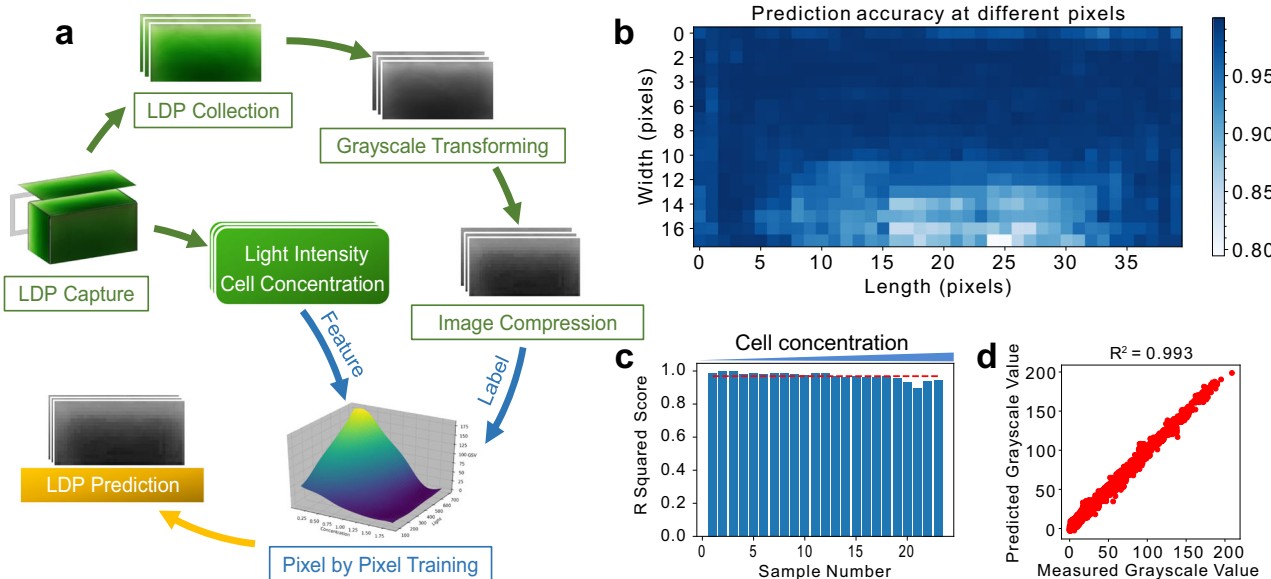

**Fig. 1 Data processing and machine learning.** Data pre-processing (Green arrows), machine learning training (Blue), and prediction process (Orange) are shown in **a**. Light distribution patterns (LDPs) inside a PBR with varied cell concentrations under different light intensities are captured and transformed to grayscale images, followed by compression to 40 × 18 pixels. The light intensities and cell concentrations, as well as the corresponding 40 × 18-pixel LDPs, are used as features and labels, respectively, in the machine learning training. In order to achieve accurate prediction, the training and prediction are performed pixel by pixel. **b** Pixel-by-pixel $R^2$ evaluation of LDPM prediction over testing samples suggests LDPM performs well at the majority of pixels. Evaluation over all pixels on testing LDPs showed an $R^2$ score of 0.993 (**d**), further verifying the accuracy of the LDPM. **c** linear regression shows near-linear correlation between GSV and light intensity across all cell concentrations (average $R^2$ score at 0.969), suggesting the grayscale value is a legitimate representation of light intensity. Cell concentrations from left to right are 0.11973, 0.21294, 0.40872, 0.45162, 0.54405, 0.62712, 0.74256, 0.82056, 0.90948, 0.96915, 1.10604, 1.2246, 1.3026, 1.3923, 1.443, 1.5444, 1.7901, 1.9188, 2.0241, 2.3556, 2.535, 2.9601, and 3.6777 g/L. Source data are provided as a Source Data file.

a second machine learning model to predict growth rates based on LDPs. We named this second machine learning model a growth rate prediction model (GRM).

The overall workflow for GRM training is shown in Fig. 2b. Vectors extracted from LDPs and their corresponding growth rates (based on the same time points) were set as features and labels in the training, respectively (See details in Supplementary Method 4 and 5). As shown in Fig. 2c, the validation rendered an $R^2$ value of 0.992, verifying the accuracy of GRM prediction. The results established quantitative connection between light availability and cell growth rates. The success in growth prediction indicates that machine learning could be introduced as an effective tool to monitor or stimulate algal growth, inform light management, and guide cultivation system design.

**Machine learning-informed semi-continuous algal cultivation sustains high biomass productivity.** The ability to predict algal growth is critical to algal cultivation management and design. For example, given light conditions over the coming days and current cell concentrations, growth prediction could indicate the optimal harvest time and how much to harvest for maximum productivity and profit. Empowered by machine learning models, we were able to simulate cyanobacterial growth under different constant light conditions by combining the LDPM and GRM (Fig. 2a, See details in Supplementary Method 6). As shown in Supplementary Fig. 4a–f, the simulated growth was very close to measured growth at all tested conditions, with a lowest $R^2$ value of 0.996. We also tested if the machine learning models could simulate cyanobacterial growth under changing light conditions. As shown in Supplementary Fig. 4g, growth predictions under changing light achieved an $R^2$ score of 0.978 compared to measured results, validating the accuracy of the model. Overall, the results demonstrated that machine learning models could accurately

simulate cyanobacterial cell growth at both constant and changing light conditions. The machine learning model is thus more versatile compared to traditional mathematical models (e.g., models based on the Monod equation) and does not require prior knowledge of growth characteristics. Moreover, the machine learning-based growth simulation is highly flexible and could expand to integrate other growth impacting factors such as temperature and nutrients. Such integration might be too complicated for traditional mathematical models, especially under changing light.

Growth simulation could inform cyanobacterial cultivation to overcome mutual shading. Although many strategies (e.g., illumination optimization, increasing bubbling rates) have been proposed to overcome light limitation, their productivity improvements were limited and not sustainable[21,23,24]. Empowered by growth prediction, we propose a type of algal cultivation system where cells are removed periodically or continuously to maintain the cultivation with near-optimal light availability and growth rates. The continuous or semi-continuous cultivation systems could minimize the impact of mutual shading and improve growth potential for cyanobacteria. As a demonstration, we simplified the SAC system with a harvesting interval of 24 h and used machine learning-based growth simulations to predict the best initial inoculum concentration. We evaluated biomass productivity predictions from different initial cell concentrations under low light (107 μmol m$^{-2}$ s$^{-1}$), high light (714 μmol m$^{-2}$ s$^{-1}$), and changing light (178-714-178 μmol m$^{-2}$ s$^{-1}$). As shown in Fig. 2e–g, the simulated productivities showed similar trends to measured productivities at all tested light conditions. Measured productivities from constant light conditions were very close to predicted productivities (Fig. 2e–g), while minor deviations were observed under changing light (Fig. 2g). The deviation could have resulted from slower growth due to adaptation to light changes. Overall, the results reveal the effectiveness of machine learning-based

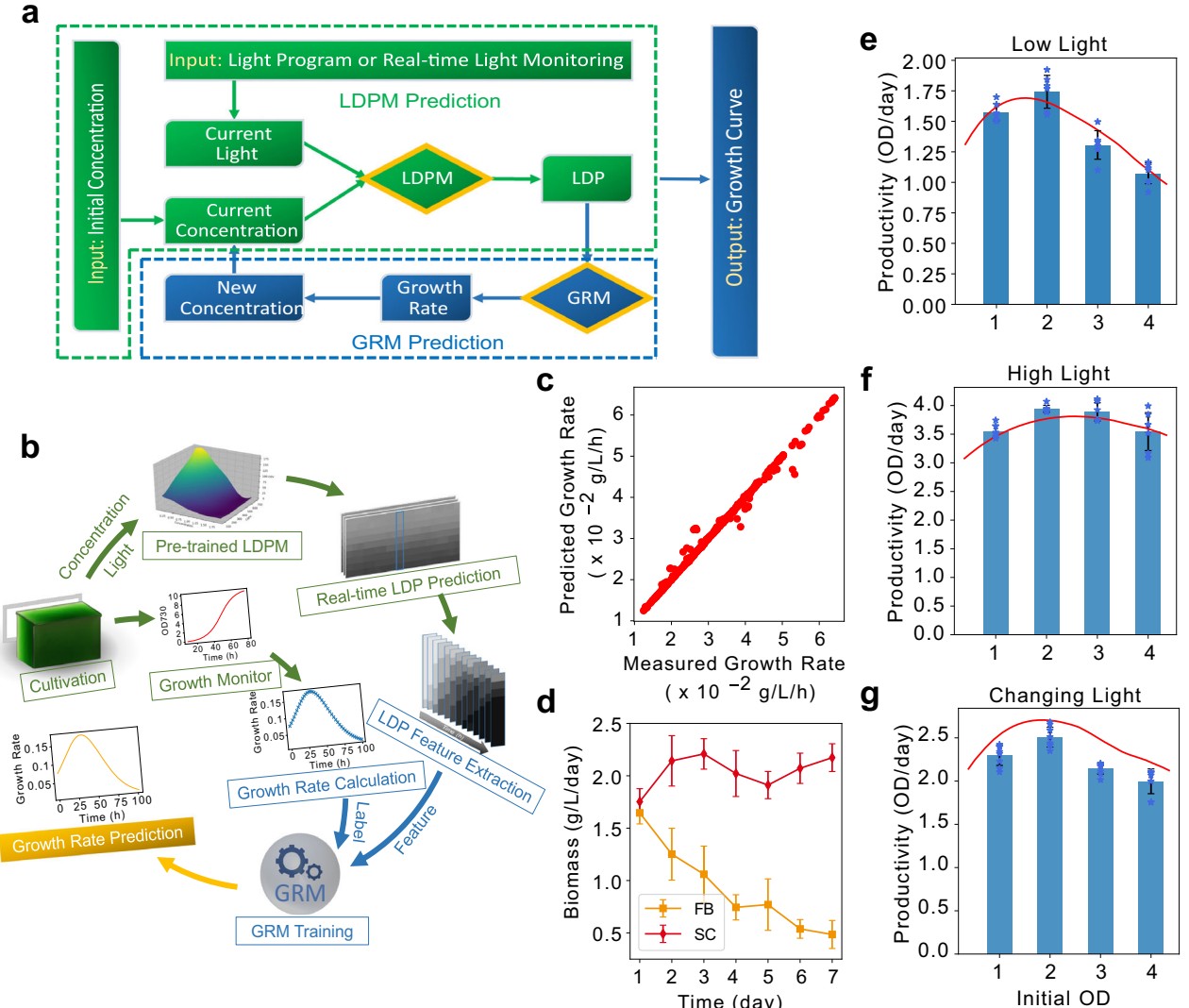

**Fig. 2 Growth rate prediction, growth simulation and semi-continuous algal cultivation (SAC).** Overview of workflows for growth simulation and GRM training are shown in a and b, respectively. The growth simulation could be achieved by integrating the LDPM (Green) with an additional growth rate prediction model, GRM (**a**, Blue). The LDP features predicted by LDPM and corresponding growth rates calculated from growth curves were used as features and labels, respectively, in the GRM training (**b**). The accuracy of GRM prediction was evaluated and shown in **c**, with an $R^2$ score of 0.992, indicating high precision. The cyanobacterial growth (biomass production) with different initial OD under low light (**e**), high light (**f**), and changing light (**g**) were simulated (red lines) and monitored (blue columns). The similar trends between simulations and measurements verified the accuracy of the simulation and legitimized the simulation as a reliable tool to inform algal cultivation system development. After light condition optimization, biomass productivity from machine learning-informed SAC was evaluated, with fed-batch as control. The biomass productivities sustain at around 2 g/L/day over 7 days in SAC (**d**, SC) while they decreased over time in fed-batch (**d**, FB). Original data points of bar figures (**e**–**g**) are shown on the plot with blue stars. Data are presented as mean values ±standard deviations ($n = 3$ independent samples with three technical replicates). Source data are provided as a Source Data file.

growth simulation in guiding cultivation platform advancement. In real-world applications, in addition to predicting optimal initial cell concentration, growth simulation could determine when and how much algal biomass to harvest under certain growth conditions. The prediction could be used in combination with economic analysis for maximized profits.

Despite higher biomass productivity using optimal initial cell concentrations in SAC, the growth rate of UTEX 2973 was less than previously reported[25–27]. In order to further improve biomass productivity, we optimized light conditions with double light sources at 574 μmol m$^{-2}$ s$^{-1}$ on opposite sides of PBRs. To determine the best initial cell concentration for the updated SAC, we adapted the machine learning models for double-light growth simulation. The prediction suggested that OD$_{730}$ ~2.3 is the

optimal initial cell concentration for SAC (Supplementary Fig. 5). Thereafter, we set up the SAC under double light sources at 574 μmol m$^{-2}$ s$^{-1}$ and maintain the initial OD$_{730}$ at ~2.3 after each harvest to allow the cells to grow back from an optimal starting concentration.

Cyanobacterial biomass productivities in SAC were evaluated with fed-batch cultivation (FB) as a control. The growth of cyanobacteria in fed-batch and SAC is shown in Supplementary Fig. 4h. As shown in Fig. 2d, biomass productivities in SAC were maintained at ~2.0 g/L/day over 7 days, while productivity in fed-batch cultivation decreased to 0.4 g/L/day on day 7 (Fig. 2d). The results suggest that machine learning-informed SAC effectively overcomes growth limitations caused by mutual shading and significantly improves and sustains biomass productivity. Such

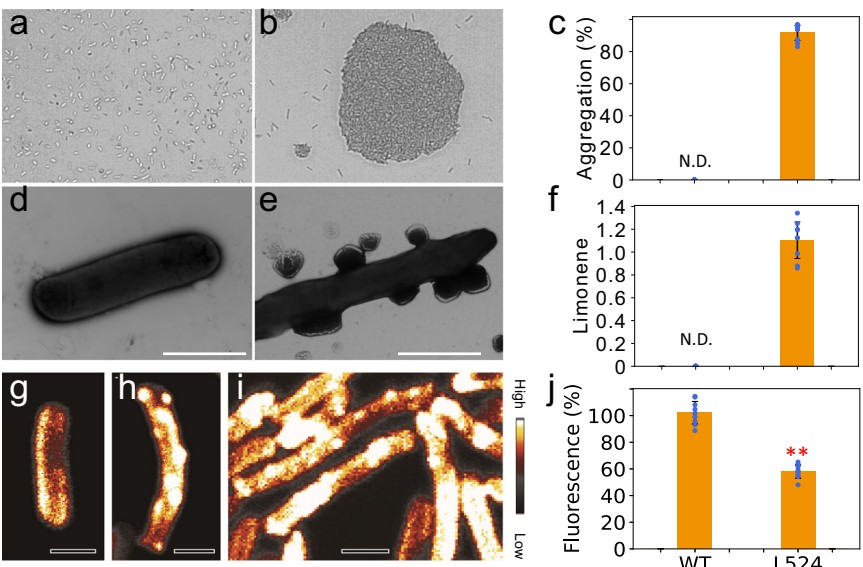

**Fig. 3 Limonene production enables cell aggregation in UTEX 2973.** Cell aggregation is observed in L524 (**b**) but not in wild-type (WT, **a**). Quantification analysis suggests 91% of L524 cells are found in aggregates (**c**). Such aggregation could result from limonene production in L524. Indeed, putative limonene droplets are found on L524 cells (**e**) but not in wild-type (**d**) as shown by TEM images, and limonene production is only detected in L524 at ~1.4 mg/L/day/$OD_{730}$ in normalized productivity by GC-MS (**f**). SRS chemical imaging was used to identify chemical compositions in the droplets. A significantly higher limonene signal is found in L524 (**h**) compared to wild-type (**g**). This observation is more evident at the L524 cell surface, where limonene droplets appear to attach to the outer cell surface. In L524 cell aggregation, the surface-attaching limonene appears to form inter-cell junctions bridging cells (**i**). Moreover, cell surface hydrophobicity was measured by BATH assay. The significantly reduced fluorescence signal in L524 (**j**) suggests over 40% of L524 cells bind to hydrophobic hydrocarbon, confirming their increased cell surface hydrophobicity. Scale bar, 2 μm. N.D. not detected. **$p = 8.4 \times 10^{-10}$ (two-tailed Student's t test, n = 3 independent samples with three technical replicates). Original data points of bar figures (**c**, **f**, and **j**) are shown on the plot with blue dots. Data are presented as mean values ±standard deviations (n = 3 independent samples with three technical replicates). Source data are provided as a Source Data file.

success could encourage further development in artificial intelligence to guide algal cultivation system design, refine cultivation management, and automate process operation.

**Altering cell surface hydrophobicity to achieve efficient cell aggregation.** Despite the potential of SAC, its feasibility depends heavily on cost-effective harvesting, a major challenge in algal biofuel. Sedimentation or auto-flocculation represents an ideal method for cyanobacterial biomass harvesting[3–5], but auto-flocculation and sedimentation without chemical or micro-organism additions remain challenging for single-cell algae. According to Stokes' Law, sedimentation rate is determined by the size and density of particles[3]. UTEX 2973 cells contain around 42.8% protein, 36.5% carbohydrates, and only 11.2 % lipid. Due to the high carbohydrate content (average density ~1500 kg/m$^3$), high protein content (average density around 1300 kg/m$^3$), and low lipid content (average density around 860 kg/m$^3$) of UTEX 2973 cells[3], they should be dense enough for sedimentation in water (~1000 kg/m$^3$). We suspected that auto-flocculation or sedimentation of UTEX 2973 could be achieved by increasing particle size via cell aggregation.

One approach to achieve cell aggregation is to increase cell surface hydrophobicity to promote cell-to-cell self-adhesion[28]. We hypothesized that engineering hydrophobic molecule production could increase cell hydrophobicity and drive cell aggregation for sedimentation. To test this hypothesis, we overexpressed a limonene synthase in UTEX 2973 to produce limonene, a strong hydrophobic terpene that can be excreted from cyanobacterial cells[29–31]. The strain was named L524. A cell aggregation study showed that aggregation occurred in L524 (Fig. 3b), but not in the wild-type (Fig. 3a). Quantitative analysis

demonstrated that 91% of L524 cells aggregated after 30 min (Fig. 3c).

To further understand if the aggregation resulted from limonene production, we observed L524 cells under Transmission Electron Microscopy (TEM) and verified the limonene production by gas chromatography–mass spectrometry (GC-MS). Putative limonene droplets were found on L524 cells (Fig. 3e) but not on wild-type cells (Fig. 3d). The formation of the droplets might be a process for limonene to secrete from cells. Indeed, limonene production was detected by GC-MS in L524 at ~1.4 mg/L/day/$OD_{730}$ (Fig. 3f).

To further verify the accumulation of limonene in L524 cells, stimulated Raman scattering (SRS) microscopy was used to visualize limonene distribution in cyanobacterial cells[32,33]. As shown in Fig. 3g–i, the weak signal from wild-type cells (Fig. 3g) can be considered background since limonene production was not detected by GC-MS in the wild-type (Fig. 3f). By contrast, strong limonene signals were observed in L524 cells, primarily presenting as droplets (Fig. 3h). These results support the hypothesis that droplets found on the L524 cell surface by TEM were composed of limonene. More importantly, SRS imaging on L524 aggregates showed the presence of limonene at cell junctions (Fig. 3i), indicating the significant role of limonene droplets in mediating aggregation.

Limonene could promote aggregation in three ways. First, hydrophobic limonene molecules could directly increase cell surface hydrophobicity, which was supported by a bacterial adherence to hydrocarbon (BATH) assay[34]. While almost all wild-type cells stayed in the aqueous phase in the assay, over 40% L524 cells adhered to hydrocarbon as demonstrated by reduced chlorophyll fluorescence in the aqueous phase (Fig. 3j). Such hydrophobicity increases could be the driving force for cell

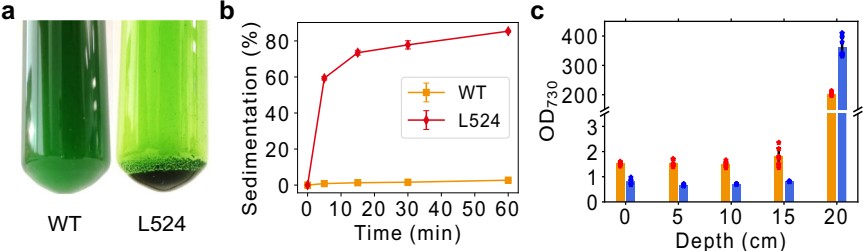

**Fig. 4 Evaluation of Aggregation-based sedimentation (ABS).** L524 cells after 1-h sedimentation are shown in **a**, with wild-type as control. By monitoring the sedimentation process, we found L524 cells started to settle within 5 min and over 75% cells were settled after 15 min (**b**), suggesting the high settling velocity of ABS. Vertical cell concentration analysis suggests that 85% and 93% cells settled to the bottom of the harvesting vessel (20 cm in depth) within 30 min and 6 h, respectively (**b**). Moreover, the biomass concentration at the bottom reaches 357 OD (139.2 g/L), which delivers 14% solids content as an output for ABS (**c**). Original data points of bar figures (**b**, **c**) are shown on the plot with red (0.5 h) or blue (6 h) stars. Data are presented as mean values ±standard deviations ($n = 3$ independent samples with two technical replicates). Source data are provided as a Source Data file.

aggregation. Second, once cells are close enough, droplets on cell surfaces could fuse to further enhance cell-to-cell adherence (Fig. 3i). Third, while a uniform negatively charged cell surface is critical to maintaining cell suspension[3,35], the neutral limonene could disrupt cell surface charge and contribute to aggregation. Moreover, the unique 'smooth' cell surface of UTEX 2973 might also promote aggregation in combination with the hydrophobic interaction of limonene production. Unlike other cyanobacteria, pili rarely form on the UTEX 2973 cell surface (Supplementary Fig. 6), presumably due to the early termination of the pilN protein[36]. The flatter cell surface of UTEX 2973 allows limonene droplets among different cells to interact with one another more easily compared to strains like PCC 7942 (Supplementary Fig. 6). Together, limonene production and the smooth cell surface might have enabled the engineered cells to aggregate due to hydrophobic interaction in a water environment.

**Aggregation-based sedimentation for efficient and cost-effective harvesting.** To investigate if limonene-induced aggregation could enable efficient UTEX 2973 cell sedimentation, we monitored the Aggregation-Based Sedimentation (ABS) process of L524 cells (Fig. 4a, b). ABS started within 5 min in L524, with over 75% of cells settled after only 15 min (Fig. 4b). A short video is provided in Supplementary Movie 1 to show the first 7 min of a mini-scale ABS. Moreover, 85% and 93% of cells settled to the bottom of the collecting vessel (20 cm in depth) within 0.5 and 6 h, respectively (Fig. 4c). The results highlight the high recovery rate and settling velocity of ABS. A major disadvantage of algal sedimentation or auto-flocculation is the low solids concentration of the output, typically between 0.5% and 3%[3]. In contrast, the cell concentration in ABS outputs reached 139.2 g/L, leading to about 14% solids content. The high solid content could result from the hydrophobic effects of limonene. More importantly, no significant differences were found between the growth of the wild-type and L524, suggesting that the limonene-induced ABS is physically prevented by air/$CO_2$ bubbling during cultivation (Supplementary Fig. 7). Overall, we demonstrated a harvest method through manipulating cell surface hydrophobicity. ABS is a cost-effective strategy with high recovery rates, sedimentation velocity, and solid content in the output. ABS could enable a sustainable and cost-effective SAC.

**Biomass and limonene yields achieved from the sustainable SAC.** Machine learning-informed SAC and ABS can be integrated for sustainable biofuel production, as shown in Fig. 5a. Besides triggering ABS for cost-effective SAC, limonene could also serve as a secondary bioproduct due to its high value and potential application in fragrance, food, and pharmaceutical

industries[37–39]. Moreover, due to its high energy density, limonene has been regarded as a 'drop-in' fuel amenable to aviation and diesel applications[29,40,41]. Thus, L524 could co-produce limonene and glycogen-rich biomass[42] from SAC. We evaluated L524 limonene and biomass productivities/yields in SAC compared to batch and fed-batch cultivations. In batch cultivation, L524 produced 11.2 mg/L limonene and 3.7 g/L biomass in 7 days (Fig. 5b, c). The limonene and biomass accumulations drastically slowed after day 2, indicating growth limitations caused by nutrient depletion (Fig. 5b, c). The limonene and biomass yields increased to 25.8 mg/L and 6.9 g/L, respectively, in 7 days with fed-batch cultivation, which removed the nutrient limitation (Fig. 5b, c). Despite the significant increases, limonene and biomass productivities still gradually decreased over time, suggesting that mutual shading became a limiting factor at high cell concentration (Fig. 5b–d). In contrast, by overcoming mutual shading, the SAC sustained near-linear limonene and biomass accumulations of ~5 mg/L/day of limonene and 2.2 g/L/day of biomass (Fig. 5b, c). The sustained high productivity resulted in 50.0 mg/L of limonene and 23.4 g/L of biomass over 11 days (Fig. 5b, c).

Limonene production by L524 from SAC surpassed previously reported yields as shown in Table 1. The high daily productivity could be attributed to the optimal light availability and the high photosynthetic capacity of UTEX 2973[26,27]. More importantly, the high yields highlight the strength of SAC in maintaining algal bioproduction at optimal rates over an extended period. A detailed comparison of productivity on the seventh day showed an ~6-fold difference in limonene productivity between SAC and batch cultivation. Similarly, Table 2 compares biomass production in relevant studies using PBRs. Although one study showed higher algal biomass productivity, the study was carried out in shaking flasks with very small volume and the addition of costly Vitamin 12 (thus not included in the comparison)[43]. We have achieved comparable biomass productivity with cultivation systems that are 20-times larger in volumes than the study. Overall, this study presented significant improvements in algal bioproduction by machine learning-informed SAC, where mutual shading has been overcome and harvesting costs substantially reduced by synthetic biology-enabled ABS.

**Scaling-up SAC with a pond system.** We further validated the potential of SAC with a 30-litre raceway pond system. We first adapted the machine learning models (LDPM and GRM) for a pond system to guide the cultivation design. Both models showed high prediction accuracy. The LDPM achieved an overall $R^2$ score of 0.986 (Fig. 6b) and pixel-by-pixel analysis suggested the LDP prediction was reasonably good at all pixels, with a minimal $R^2$

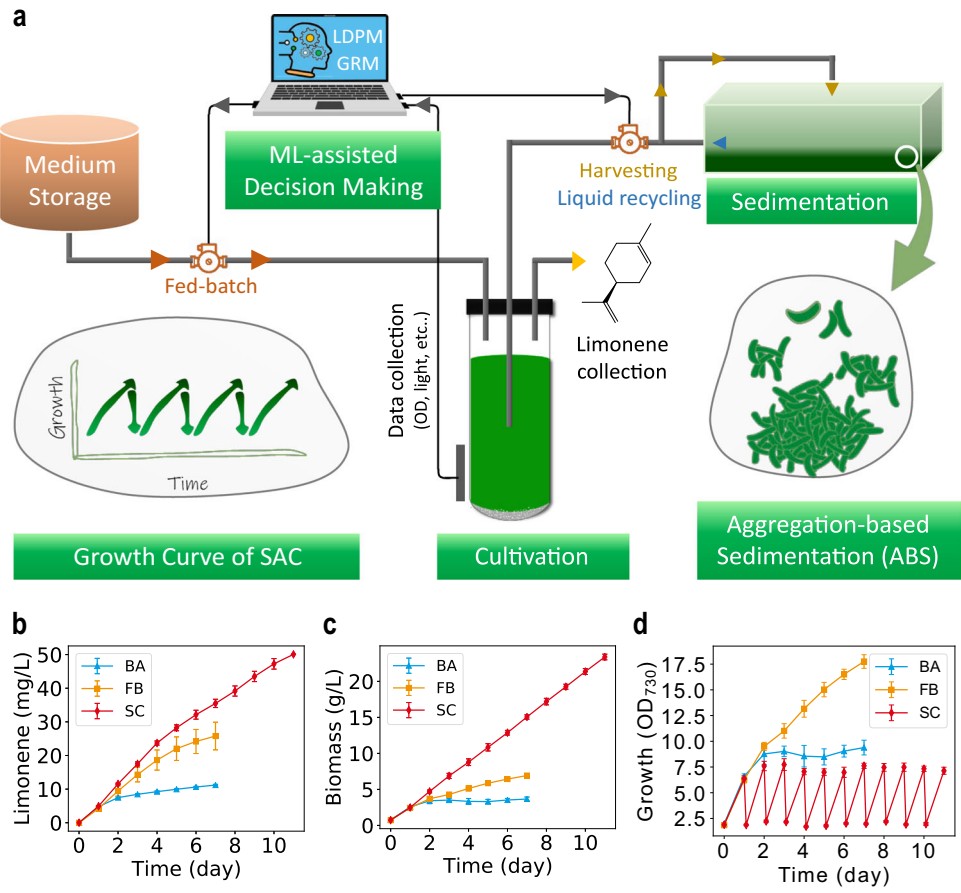

**Fig. 5 Sustainable and higher limonene and biomass productivities achieved in SAC. a** Concept figure shows the integration of machine learning-informed SAC and ABS for biofuel production. By overcoming light limitation, SAC (SC) shows sustainable limonene and biomass production, maintaining productivities at around 5 mg/L/day and about 2.2 g/L/day for limonene and biomass, respectively (SC, **b**, **c**). By contrast, both limonene and biomass production in batch cultivation (BA) reach plateau at day 2 due to nutrient depletion (BA, **b**, **c**). With fed-batch, limonene, and biomass production are enhanced but eventually flattened due to light limitation (FB, **b**, **c**). The growth of cyanobacteria with batch (BA), fed-batch (FB), and SAC (SC) are shown in **d**. Data are presented as mean values ±standard deviations ($n = 3$ independent samples). Source data are provided as a Source Data file.

**Table 1 Recent publications about limonene production in cyanobacteria.**

| Strain | Productivity (mg/L/d) | Yield (mg/L) | Time (d) | Ref. |
|---|---|---|---|---|
| *Anabaena* PCC 7120 | 0.1[a] | 0.2 | 2 | 54 |
| *Synechococcus* PCC 7942 | 0.8[a] | 2.5[a] | 5 | 29 |
| *Synechocystis* PCC 6803 | N.A. | 6.7 | 7 | 31 |
| *Synechococcus* PCC 7002 | 1.5[a] | 4 | 4 | 30 |
| *Synechococcus* UTEX 2973 | 5.0 | 50 | 11 | This study |

N.A. not applicable.
[a]The value is estimated from the figures.

**Table 2 Select publications on cyanobacterial biomass production with PBRs.**

| Strain | PBR size (cm) | Yield (g/L) | Time (d) | Ref. |
|---|---|---|---|---|
| *S. elongatus* PCC 11801 | 3[a] | 2[b] | 3 | 55 |
| *S. elongatus* PCC 11802 | 3[a] | 3[b] | 5 | 56 |
| *S. elongatus* BDU 130192 | 3[a] | 2 | 4 | 57 |
| *S. elongatus* PCC 7942 | 3[a] | 1.9[b] | 5 | 42 |
| *S. elongatus* UTEX 2973 | 3[a] | 2.3[b] | 5 | 42 |
| *S. elongatus* UTEX 2973 | 10 × 5 | 23.4 | 11 | This study |

[a]Diameter of the cylinder PBRs.
[b]The value is estimated from the figures.

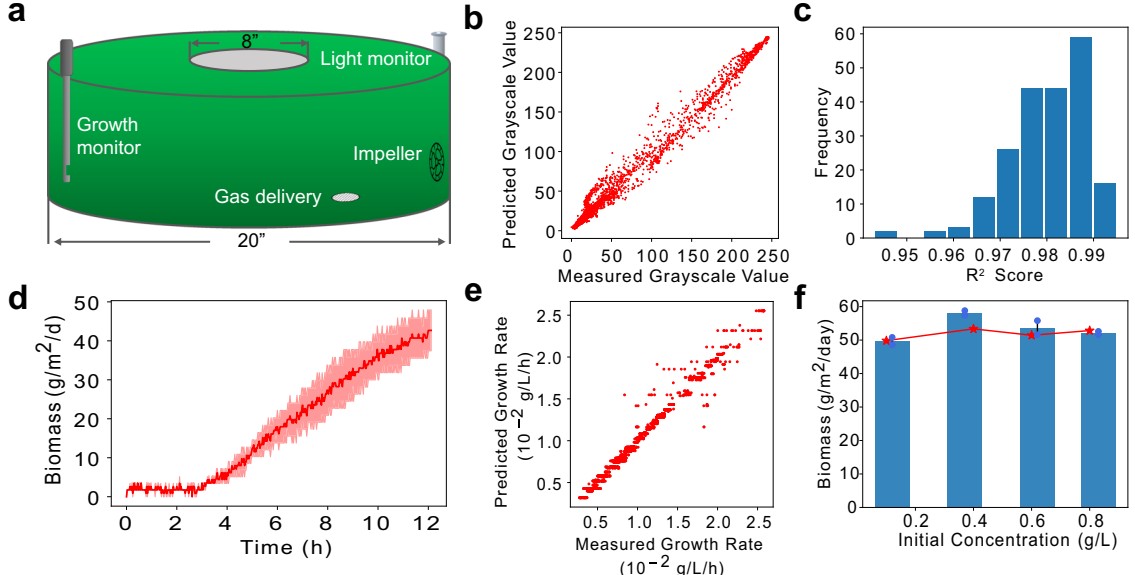

**Fig. 6 Scaling-up of SAC with a pond system.** The pond system design is shown in **a**. **b** and **c** show the prediction evaluation of the adapted LDPM for the pond system. The LDPM achieved an overall R$^2$ score of 0.986 (**b**) and pixel-by-pixel analysis suggested that prediction at all pixels were reasonably good, with a minimal R$^2$ score of 0.943 (**c**). **d**, biomass production with the outdoor pond system. e, prediction evaluation of the GRM adapted for a pond system. **f**, machine learning-based growth simulation (red line) suggested that setting the initial cell concentration to around 0.4 g/L achieves optimal biomass productivity under the growth condition mimicking Texas Summer, which was well supported by the measured results (blue bars). Original data points are shown on the plot with blue dots. Error bands represent standard deviations ($n = 2$ independent samples). Source data are provided as a Source Data file.

score of 0.943 (Fig. 6c). The GRM prediction also achieved an $R^2$ score of 0.980 (Fig. 6e). Like the PBR system, we employed the machine learning models to predict optimal initial cell concentrations for the pond SAC system. The growth simulation suggested that setting initial cell concentration to around 0.4 g/L delivers the highest biomass productivity under the growth condition mimicking Texas summer (Fig. 6f). Based on the prediction, the experimental results showed that SAC achieved the highest biomass productivity at 58.1 g/m$^2$/d (Fig. 6f). We noticed slight differences between the predicted biomass productivity and measured productivity when initial cell concentrations were around 0.4 g/L (Fig. 6f). The deviation might result from the presence of noise in the training data, and/or overfitting in the models. Future optimization such as removing noise, adding regulations, and expanding training data could further enhance the model performance. Overall, our results demonstrated the application of machine learning models in a pond SAC system. The success of application in both PBR and pond systems indicates that machine learning-based prediction can be a generalized method for guiding algal cultivation management and design in various systems.

Inspired by the high productivity from the indoor pond system, we further tested biomass productivity of the pond SAC in real outdoor conditions. The outdoor tests were carried out in late September 2021 in College Station, Texas, with both 'partially sunny' and 'mostly sunny' weather. These conditions represent a typical fall growth condition. The outdoor cultivation achieved an average biomass productivity of 43.3 g/m$^2$/d (Fig. 6d), surpassing the U.S. DOE 2022 target by 1.7 times.

**Techno-economic analysis of the pond SAC platform.** The machine learning-informed SAC holds significant economic potential after being scaled up. Recent efforts to quantify the economic potential of algal biomass production by the National Renewable Energy Laboratory (NREL) examined different existing, well-documented PBR and pond designs across a number of different configurations[44,45]. Both studies focused on estimating

the break-even minimum biomass selling price (MBSP), given an internal rate of return on capital of 10%. Based on the NREL study, the yearly average of biomass productivity is estimated to be the productivities achieved in the Spring (MAR, APR, MAY) and Fall (SEP, OCT, NOV)[44]. Following that approach, we estimated the yearly average of biomass productivity for the open pond system to be 43.3 g/m$^2$/d in the outdoor study and 48.1 g/m$^2$/d (83.3% of summer productivity) in the indoor mimicking trial. The ash content of the cyanobacterial biomass was measured to be 5.5%. At these conditions, the NREL model projects a MBSP of approximately $281 per ton based on the outdoor trial yield (Supplementary Fig. 9). By comparison, 2019 state-of-the-art open pond algal cultivation had an MBSP of ~$1,227 per ton[46]. The categorical cost distribution is shown in Supplementary Fig. 9.

Furthermore, the limonene produced by L524 has a current market value of about $5/kg[29,40]. At this price, the SAC system proposed here would generate approximately $10.08 of additional revenue in limonene sales per ton of biomass produced. Such reductions in MBSP can be readily achieved in PBR systems. Although limonene collection from open pond systems may not be cost effective at current productivity levels, limonene-mediated ABS nonetheless significantly reduces harvesting costs.

Beyond significant improvements in biomass production, the implementation of ABS in SAC would also markedly reduce operating costs. ABS (0.1 kWh m$^{-3}$) could save up to 93% on energy costs compared to traditional harvesting methods (e.g., disc stack centrifugation (1.4 kWh m$^{-3}$))[3], while maintaining high efficiency and recovery rates. As the dewatering process accounts for $24.4 per ton of biomass in the current model (Supplementary Fig. 9), the simplified harvest by ABS would further significantly reduce the MBSP (however, we have not adjusted the $281 per ton MBSP generated by the NREL model to reflect such reductions).

In addition, due to the high glycogen content of UTEX 2973 cells[42], the cyanobacterial biomass could directly feed into biorefineries for ethanol fermentation without pretreatment as

described previously[47,48]. Demand for biomass is not considered by the NREL model used here, so the additional benefit of increased willingness-to-pay for biomass from the L524 and SAC platform is not quantified. While still in the early stages of development, the SAC platform with the L524 strain appears to overcome many of the challenges that have long plagued algal biofuel production.

Together, significant increases in algal productivity and reductions in operating costs result in a dramatic reduction in the break-even biomass price relative to prior algal production systems to below $300 per ton of AFDW. Detailed work must be done to provide robust cost estimates, but the initial results show great promise. At the same time, the SAC process would generate biomass that is significantly less costly to convert to ethanol than the current most common feedstock (corn), as it would eliminate the need for costly milling and other pre-treatment prior to fermentation[47,48].

## Discussion

The research has led to several breakthroughs that could have a profound impact on biomanufacturing, algal bioproduction, and renewable fuels and products. First, the study is one of the initiatory to use Artificial Intelligence techniques to guide algal cultivation design. In particular, the research provided quantitative insights into how light intensities and cell density shape LDPs and how LDPs, in turn, impact cyanobacterial growth rates. The integration of LDPM and GRM enables reliable simulation of growth curves based on initial OD and light intensity. This knowledge inspired us to develop SAC and precisely define the optimal initial OD to achieve maximized growth. The high accuracy, broad prediction range, and superior capacity to handle the complexity of machine learning models produced broader adaptability in constant or changing light and in indoor/outdoor PBRs or pond systems. The principle and design of the study can be broadly applied to industrial microbiology and biomanufacturing. The machine learning models themselves can be broadly adapted to different set-ups to guide algal cultivation management and design. The models can be further optimized to integrate nutrients, temperature, and other factors to achieve even broader adaptability.

Second, the study achieves aggregation-enabled sedimentation (ABS) by manipulating cyanobacterial cell hydrophobicity. Self-sedimentation achieved a high solids load and enabled an efficient and low-cost harvest method for algal bioproduction, overcoming a major challenge in the algal industry. Furthermore, the principle can be used to design ABS in other species for broader biomanufacturing applications.

Third, the study achieved increased yields of biomass, in both indoor and outdoor systems, in both PBR and pond systems. The outdoor raceway pond productivity achieved 43.3 g/m²/d, which surpasses the U.S. DOE 2022 target by 1.7 times. The consistency of outdoor productivity and indoor estimated productivity (43.3 g/m²/d vs. 48.1 g/m²/d) again proves the effectiveness and reliability of the approach in the study. Due to enhanced yields and reduced operating costs by ABS, SAC holds great promise for economical algal bioproduction below $300 per ton. Furthermore, the lower cost of algal biomass enables economically competitive applications in broader industries, including algal biofuel, animal feed, food additives, and various speciality products[47–50].

## Methods

**Strains and growth condition**. *S. elongatus* UTEX 2973 wild-type was kindly gifted by Dr. Pakrasi from Washington University. Strains were maintained in BG11 (Sigma, C3061) supplemented with 10 mM TES under 50 µmol photons m⁻² s⁻¹ illumination at 37 °C. A customized PBR (based on a 1-L Roux bottle) containing 500 ml of media was used for cultivation, with 5% (vol/vol) $CO_2$ bubbling from a stainless-steel aeration stone at a speed of 0.8 L/min. 10 ml of 50× stock media was fed every 24 h for fed-batch cultivation. For SAC, cell concentration was adjusted to $OD_{730}$ of ~2.3 every 24 h followed by media feeding. The initial OD was selected based on the machine learning model outcome of optimal starting OD. The growth temperature of batch cultivation, fed-batch, and SAC was maintained at 37 °C. Artificial light at 574 µmol m⁻² s⁻¹ was applied on two opposite sides of the PBR, after initial growth with one-side 357 µmol m⁻² s⁻¹ and 714 µmol m⁻² s⁻¹ at 0–12 h and 12–36 h, respectively. A customized pond system was used for scaling-up of the SAC, shown in Fig. 6a. The circular pond system contained a 6-inch-wide raceway and an impeller was used to keep the cyanobacterial cells agitated. In all, 30 litres of cyanobacteria (20 cm in height) were cultivated in the pond system with 5% $CO_2$ (vol/vol) bubbling via gas dispersion stones. The growth temperature was maintained at 40 °C with a water heater. Cell growth and light conditions were monitored with a turbidity meter (EXcell231, EXNER, with Expert software) and a light sensor (LS-BTA, Vernier, with Vernier Graphical Analysis software), respectively. In the condition mimicking Texas summer, the pond system was placed in a growth chamber and the light program was set to 400 µmol m⁻² s⁻¹ for 1 h, 800 µmol m⁻² s⁻¹ for 1 h, 1300 µmol m⁻² s⁻¹ for 1 h, 1500 µmol m⁻² s⁻¹ for 10 h, 1300 µmol m⁻² s⁻¹ for 1 h, 800 µmol m⁻² s⁻¹ for 1 h, and 400 µmol m⁻² s⁻¹ for 1 h (all light intensities were measured from the pond surface). In both outdoor and mimicking outdoor conditions, 250 ml water was added to the pond system every 2 h to counter evaporation.

**Molecular manipulation of cyanobacteria**. A construct, pLB524, was used to create the strain L524 via homologous recombination. To build pLB524, homologous sequences of UTEX 2973 neutral site I and limonene synthase were amplified from pWX1118[29] with primer pairs of NS-DS-F/ NS-US-R (Supplementary Table 1). The amplified fragment was then integrated into pBR322 by Gibson assembly. The assembled pLB524 was transformed into UTEX 2973 by conjugation[25,36]. Briefly, cargo *E. coli* strain containing pLB524 and helper plasmid pRL623 was first mixed with a conjugal strain containing pRL443 for 30 min at 37 °C, before mixing with UTEX 2973 cells. The mixture was then incubated on BG11 + 5% LB plates without antibiotics and then transferred to BG11 plates with 5 µg/ml spectinomycin/streptomycin. Transformants that had been segregated with increasing antibiotics (5 µg/ml, 10 µg/ml, and 15 µg/ml) for three rounds were verified by PCR and further confirmed by qPCR with primers provided in Supplementary Table 1.

**Microscopy imaging and aggregation evaluation**. Cells sampled from cyanobacterial culture were adjusted to the same concentrations and transferred to Eppendorf tubes for aggregation. After 30 min, the tubes were gently vortexed to suspend pellets (in L524) while minimizing the perturbation for aggregation. The well-mixed samples were observed under Leica DM6B. For cell aggregation quantification, the well-mixed samples were counted with a hemocytometer. Cell aggregation was defined as aggregates with five or more cells. The number of aggregated L524 cells was estimated by subtracting the number of unaggregated L524 cells from WT cells. In the transmission electron microscopy (TEM) observation, cells were negatively stained with 1% uranyl acetate and observed under JEOL 1200.

**Chemical imaging**. SRS microscopy developed for plant biomass imaging was used to perform the chemical imaging[51]. A HighQ picoTRAIN (Spectra-Physics) laser was used to generate 1064 nm (up to 15 W) and 532 nm (up to 9 W) output; both are pulse trains at 7 ps. The 1064 nm output was used as the SRS Stokes beam. The 532 nm beam was used to pump an APE optic parametric oscillator (Levante Emerald, APE GmbH, Germany) to produce a tunable wavelength 6 ps pulse train to be used as the SRS pump beam. The 1064 nm Stokes beam was modulated by an acoustic optical modulator (3080-122, Crystal Technology) at 10 MHz frequency, achieving >80% intensity modulation depth. Both the pump and Stokes pulse trains were combined (1064dcrb, Chroma) and routed to a modified scanner (BX62WI/ FV300, Olympus) attached to an Olympus IX81 microscope. The pump beam intensity after the sample was collected by a high numeric aperture lens, filtered and detected by a photodiode. A lock-in amplifier was used to detect the stimulated Raman loss signal. The Raman frequency of the limonene C = C bond at 1670 cm⁻¹ that was previously used by other studies[33,52,53] was chosen for SRS imaging, which corresponded to a pump wavelength at 903 nm.

**Aggregation-based sedimentation measurement**. The efficiency of ABS was assessed by monitoring the sedimentation process of cyanobacterial cells ($OD_{730}$ at 10.0) in a harvesting vessel with a 20-cm height. Cell concentrations on the surface were used to evaluate the sedimentation efficiency. The vertical distribution of cyanobacteria was evaluated by sampling cells at different depths with a long glass tip.

**BATH assay for cell hydrophobicity measurement**. The bacterial adherence to hydrocarbon (BATH) assay was performed following the protocol developed by Rosenberg et al.[34] with minor modifications. Specifically, 3 ml of cyanobacteria with $OD_{730}$ of 0.2 were mixed with 0.12 ml of hexadecane. After phase separation,

the chlorophyll fluorescence of the cyanobacteria (water phase) was measured to quantify cells that did not adhere to the hydrocarbon.

**Limonene collection and measurement**. Limonene was collected with HayeSep porous polymer (Sigma) absorbent traps and eluted by 1 mL hexane supplemented with 50 µg/mL cedrene (Sigma) as the internal standard. The concentration of limonene was quantified by gas chromatography–mass spectrometry (GC-MS) (Shimadzu Scientific Instruments, Inc.) with a standard curve and normalized with recovery rates, which was determined by spiking different concentrations of limonene in 500 mL of UTEX 2973 wild-type cells (Supplementary Fig. 8). The total limonene yield was calculated by adding yields of each day together.

**Biomass productivity measurement**. The biomass productivity was measured with $OD_{730}$ and converted to dry cell weight (DCW) with a pre-established calibration (1.0 $OD_{730}$ equals approximately 0.39 g DCW $L^{-1}$). The total biomass yields were calculated by adding the productivities of each day together. The biomass productivity from the pond system was calculated by first transforming the turbidity (Attenuation Unit, AU) to $OD_{730}$ with a calibration curve (Supplementary Fig. 10b) and then calculated as described above.

**Techno-economic analysis**. The techno-economic analysis was based on the algae farm model presented by NREL[44]. Similar to the NREL study, we assumed the yearly biomass productivity to be the same as productivity achieved in Fall and set it to 43.3 g/m$^2$/d. The 50-acre individual pond size was selected for the analysis and the pond harvest concentration was set to 0.7 g/L, as the SAC output (with initial cell concentration of 0.4 g/L) was about 0.7 g/L. The primary, secondary, and tertiary dewatering outlet concentrations were set to 140 g/L, according to the ABS output concentration. We set the ash content to 5.5% and used default values for the rest of the parameters in the analysis.

**Reporting summary**. Further information on research design is available in the Nature Research Reporting Summary linked to this article.

## Data availability

Data supporting the findings of this work are available within the paper and its Supplementary Information files. A reporting summary for this article is available as a Supplementary Information file. Training data for machine learning models are available at GitHub [https://github.com/joshuayuanlab151/LDPM-and-GRM]. Source data are provided with this paper.

## Code availability

Code used for machine learning models and training data is available at GitHub [https://github.com/joshuayuanlab151/LDPM-and-GRM]. A stable release is available at Zenodo [https://zenodo.org/badge/latestdoi/430008654].

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

## Acknowledgements

The authors would like to thank Dr. Himadri Pakrasi from the Department of Biology, Washington University in St. Louis for kindly gifting the UTEX 2973 strain, and Dr. Stanislav Vitha and Dr. Rick Littleton from the Microscopy Image Center (MIC), Texas A&M University for their assistance with microscopy imaging. The authors also acknowledge the funding supports of Dr. John Hood's donation (Hood Fund for Sustainability), Texas A&M AgriLife's Chair Funds for Synthetic Biology and Renewable Products to J.S.Y., and Research and Development Fund from Texas A&M University. The on-going research is supported by the DOE Fossil Energy Office (DE-FE0032108). Yining Zeng acknowledges the support from the Laboratory Directed Research and Development (LDRD) Program at NREL. B.L. and M.L. acknowledge the scholarship from the China Scholarship Council.

## Author contributions

B.L. and J.S.Y. designed the experiments. B.L. carried out the machine learning, modeling, simulation, and strain engineering. B.L. and M.L. did the strain cultivation and condition optimization. Y.Z. performed the SRS chemical imaging analysis. Z.A., B.F. and H.B. performed the TEA presented in this paper. Q.L. performed the ash content measurements. B.L., S.Y.D. and J.S.Y. discussed the results and co-wrote the manuscript. All authors contributed to the scientific discussions and comments on the manuscript.

## Competing interests

The Texas A&M University System has filed a patent application (application number: 63/299,162, inventor: B.L. and J.S.Y.) on the machine learning informed cultivation and the limonene enabled sedimentation presented in this work. Other authors claim no competing interest.
