## [Peer Review File · Nature Communications]

Machine Learning-Informed and Synthetic Biology-Enabled Semi-Continuous Algal Cultivation to Unleash Renewable Fuel ProductivityReviewers' Comments:

Reviewer #1:

Remarks to the Author:

The authors claimed two independent works of machine-learning based light distribution pattern and development of a self-flocculation.

- Authors developed that the LDP is useful to quantify the mutual shading. The GRM predicted the growth productivity depending on the light sources, which were used to harvest at the right time as a SAC. I wonder how the ML-assisted model improved the model prediction, which was not fully described. Besides, this tool limits indoor cultures with artificial light sources. This cannot be generalized in the outdoor culture, which is a more proper or feasible culture for massive cultivation so far.
- The model suggested that light is a key factor for cell growth, which was not fully demonstrated or described. The authors simply explained the Fig. 2D-F that they were able to predict the productivity based on initial OD and light. I am not sure that how this model is superior to a Monod type equation in general at a fixed light source. This point, comprehensive optimization must be followed to support the predictive model, which lacks in this study.
- There was limited information about the SAC system given in this manuscript. Thus, it cannot be evaluated for the performance and feasible operation. This also limits understanding of how the ML assists the SAC. Besides, no applications were performed using the SAC., except the limonene production below. An independent application must be followed to demonstrate the usefulness. In Fig. 2G, biomass was compared to the FB as a control, in terms of productivity (g/L/d). No sufficient data are given (i.e. growth, CO₂ uptake or conc., Chl a, etc over time).
- Authors claimed that engineered strain exhibits increased hydrophobicity by producing limonene. Then, the droplets were analyzed via several tools including TEM, SRS and, BATH. Previously, limonene was volatile and their collections from the medium have been studied. However, this study showed that hydrophobic products were found on the surface, acting as an aggregating agent. Although various analyses have demonstrated the hydrophobicity of limonene-producing cells, the SAC culture was not fully described for this reviewer to evaluate the novelty and impact of the system.
- Authors claimed that the SAC platform is economically feasible. This evaluation was based on the indoor culture with limited scales. Thus, it is not proper to evaluate the TEA.
- Base on the limitation in this study, I do not recommend publication in the journal.

Reviewer #2:

Remarks to the Author:

The manuscript entitled "Machine Learning-Informed and Synthetic Biology-Enabled Semi-Continuous Algal Cultivation (SAC) to Unleash Renewable Fuel Productivity" applied ML methods to predict light spatial distribution and the microalgal growth rates. Also, they designed a low-cost biomass harvesting process, which is one of the main challenges for microalgal production. In my opinion, this study is within the scope of Nature Communications. It is a very interesting paper, and I recommend its publication after minor changes. Please see my comments below.

Minor comments:

- L81: How were greyscale values calculated? Was the light intensity measured at the surface of PBR (and in the two dimensions)?
- L93: How were these percentages calculated? R-squared values are calculated for a set of values. Do

you divide the data to determine different R-squared values?

- L112: How was the correspondence between LDP and specific growth rates attributed?
- L175: And the density of the lipids? What is the mass fraction of lipids on biomass?

Reviewer #3:

Remarks to the Author:

This study investigated using machine learning models to predict light availability and algae cells growth rates, which enables the design of Semi-continuous Algal Cultivation (SAC) to achieve high algae productivity. In addition, to realize low cost algae harvesting under SAC mode, aggregation-based sedimentation (ABS) was developed via engineering a fast-growing strain to generate limonene to enables efficient cell aggregation and sedimentation. The combination of SAC and ABS via engineering strains to generate limonene is very interesting and innovative. This method can greatly reduce the algae cultivation cost via high productivity and low cost harvesting. The research methods used in the study well supported major results and conclusions, except for the cost analysis part. Therefore, the manuscript needs modifications before being considered for publication.

Major comments are:

1. Line 158 to 159, as claimed by the authors, "we set up the SAC under double light sources", this SAC mode based on the machine-learning model developed in this study primarily can only be used for photobioreactor (PBR) with double light sources. This limited the application of this machine-learning model to other types of algae cultivation systems, such as open pond and single-slight source PBR.
2. Line 165 to 168: First, this is not "the first reported machine-learning-informed algal cultivation design." The authors should do more literature searches. Although very limited, at least one report has been published about using machine-learning models to guide algal cultivation design based on light distribution. Please modify this claim. Also together with comment 1, the application of this machine-learning models with double light sources is limited considering various algae cultivation methods. Please discuss the possible improvement that can be done for the machine-learning models.
3. Line 175 to 178: This sentence is awkward and confusing. High enough density is typically needed for sedimentation no matter what carbohydrates and proteins contents of the algae. Please rewrite this sentence and clearly describe the carbohydrates/proteins contents impacts on cell densities.
4. Line 227: This is a big achievement. If the ABS output cells concentration can reach 139.2 g/L, it will greatly reduce the harvesting cost by lower dewatering cost and energy needs.
5. Line 229 to 231: this conclusion might need a longer testing time to be verified, such as 10 days or 1 month. As air or CO₂ bubbling is not always on, there might be some aggregation happened during the growth phase compared to the wild type. If so, it will affect the light adsorption of the algae cells.
6. Line 252: Is the productivity dry basis or dry ash free basis? If dry basis, please provide dry ash free basis or ash % information.
7. Line 281 to 282: The "40 g/m²/d" productivity is different from my calculation results based on the "0.1 g/L/h" in your study and information in ref 45. Please explain how the 0.1 g/L/h is translated into 40 g/m²/d.
8. Line 285: Please define or provide the major cost parameters for this "break-even MBSP" estimation. For example, what is the internal rate of return for this "break-even" MBSP, 0%? In addition, to translate to a horizontal tube PBR system, what is the productivity for the MBSP estimation?
9. Line 299 to 300: "quite possibly to below \$300 per ton AFDW." This conclusion cannot be clearly obtained from the data or inputs provided by the authors. At least, some major cost assumptions directly related to this conclusion should be clearly provided and described. For example, what is the PBR system assumed for this MBSP? What is the productivity? What are the reference values (productivity and MBSP) used as a starting point or basis for the result of \$300/ton AFDW? Are there other major cost assumptions used here? No robust cost estimates is requested, but the authors should clearly demonstrate the cost analysis basis for the \$300/ton AFDW MBSP result.

10. Figure 1 and 2 have some repetitive information. Figure 1 A and figure 2 A should be combined into a single figure. Figure 1 C, D and E seems redundant, which are suggested to use only one figure to represent the R-squared scores for pixels prediction.

Minor comments are:

1. Line 285: "of" should be "to".

Authors' response to the reviewers:

Comments from Reviewer 1

The authors claimed two independent works of machine-learning based light distribution pattern and development of a self-flocculation.

Response:

We appreciate the comments. In fact, the outcomes of machine-learning models and aggregation-based sedimentation (ABS) were well integrated to enable a unique the semi-continuous algal cultivation (SAC), which delivered record biomass productivity in both pond and PBR system, along with the record limonene titer. The machine-learning models informed the necessity for SAC, established the concept, and identified the optimal starting OD and the growth rate for SAC. The ABS enabled frequent low-cost harvest, making SAC practical and economic. TEA was carried out to evaluate the economics of this new platform. The manuscript presented a perfect path from computational modeling to experimental implementation and system evaluation. The two technologies were well integrated to achieve multi-level of novelty: the novelty for machine learning model application, the novelty for understanding and light penetration and growth impact, the novelty for SAC to overcome light penetration limitations, the novelty for unique synthetic biology design to enable cell aggregation, the novelty for efficient self-sedimentation with high solid content, and the novelty of practical and economic SAC and algal biomass production.

Comment-1a: • Authors developed that the LDP is useful to quantify the mutual shading. The GRM predicted the growth productivity depending on the light sources, which were used to harvest at the right time as a SAC. I wonder how the ML-assisted model improved the model prediction, which was not fully described.

Response:

We appreciate the comments about the usefulness of LDP and strength of the machine learning-based growth predictions. In fact, none of the previous work was able to integrate the LDP prediction with growth prediction. Such novelty informed the design for SAC. We also appreciate the comments to ask us to fully describe 'how the ML-assisted model improved the model prediction'. We hereby summarize four advantages of ML-based modeling: the preciseness, the range with accurate prediction, the two-dimensional capacity, and the flexibility and better integration with growth models. These strengths enabled accurate growth model to be integrated with LDP.

- 1. ML-based modeling is empirical and can achieve very high accuracy. With the reviewer's suggestion, we further optimized the algorithm and achieved R^2 of 0.995. Recent mathematical model based on Beer-Lambert's law for light penetration prediction has achieved R^2 of 0.98¹. The accuracy ML-model is superior as compared to other models.*
- 2. The machine learning models can be used in a much larger range of cell concentration. With cell concentration increasing, the light distribution becomes more complex due to light scattering and interferences. The cell concentration for LDP prediction with traditional mathematical methods were normally limited below 1 g/L¹⁻³. However, the LDPM presented in this study can accurately predict LDPs with cell concentration from 0.1 g to 3.9 g/L. The capacity is important to generate output to allow accurate growth*

prediction.

- 3. The LDP can be multi-dimensional with machine learning, while it is limited in one-dimensional in traditional mathematical methods. In a real-world setting, LDPs are usually two or three dimensional. The multi-dimensional LDPs predicted by machine learning models are more representative for real-world light availability in algal culture.*
- 4. The LDPM is more flexible. Because the model is based on empirical training, it can be applied into any existing cultivation system for any conditions to explore. Unlike mathematical modeling, it does not need to have a defined understanding of the characteristics of the response.*

With all these aspects being said, the better accuracy, larger range, and superior capacity and flexibility all enabled us to integrate the outcome of predicted LDPM (LDP model) into growth model to allow accurate growth curve prediction, guiding SAC design. This would be challenging if using traditional models to predict light penetration. We hereby expanded the discussion of ML-based LDP prediction as follows in line 103-116. ‘The high R-squared score (0.995) highlights the increased accuracy of the machine learning model over traditional mathematical models. Furthermore, unlike mathematical models that can only predict one-dimensional light paths, machine learning-predicted LDPs can be two-dimensional or even three dimensional. Moreover, the upper cell concentration limit of the LDPM is about 3.9 g/L, which is far higher than the limit of ~1 g/L presented in previous mathematical models¹⁻³. The larger prediction range indicates that a machine learning-based strategy could address LDP prediction challenges caused by complex light scattering and interference at high cell concentrations. The methodology for LDP prediction proposed in this study could be transferred to any existing algal cultivation systems, such as indoor/outdoor PBRs or pond systems. The superior performance of the machine learning model—in particular, a larger prediction range and higher accuracy—enabled LDP outputs to be used to simulate growth curves using a second machine learning model. Such integration has not been achieved in previous studies and would guide cultivation optimization.’

Besides the ML strength in LDPM and integration of LDPM and GRM, the strength of ML for growth simulation includes:

- 1. The machine learning-based growth prediction can be used to predict growth under both fixed light condition and changing light conditions, while traditional methods limited in specific growth conditions. This is a significant limitation for conventional method, as algal outdoor growth will involve changing light.*
- 2. The machine learning-based method is more flexible, and can be expanded by integrating other growth factor such as temperatures and nutrients. Such integration might introduce too much complexity to allow accurate prediction of mathematical models. As compared to Monod model, the machine learning model has achieved accurate prediction of cell growth among all growth phases, under complex situations.*
- 3. Last but not least, mathematical modeling requires defined factors and functional understanding for equation development. However, machine learning model is empirical and can integrate the factors to achieve accurate*

modeling without defined understanding.

In terms of the growth prediction, we discussed the advantages of machine learning-based method by adding sentences ‘Overall, the results demonstrated that machine learning models could accurately simulate cyanobacterial cell growth at both constant and changing light conditions. The machine learning model is thus more versatile compared to traditional mathematical models (e.g., models based on the Monod equation) and does not require prior knowledge of growth characteristics. Moreover, the machine learning-based growth simulation is highly flexible and could expand to integrate other growth impacting factors such as temperature and nutrients. Such integration might be too complicated for traditional mathematical models, especially under changing light.’ in line 153-161.

Because of the advantages, we believe the machine learning-based prediction is much better than mathematical models in guiding the algal cultivation system design. We have discussed the above superiority separately in the main text.

Comment-1b: Besides, this tool limits indoor cultures with artificial light sources. This cannot be generalized in the outdoor culture, which is a more proper or feasible culture for massive cultivation so far.

Response:

We appreciate the comments very much. The comment actually stimulated us to carry out experimental and modeling work to demonstrate the broader applicability of the machine learning models. In fact, the machine learning-based tool is not limited by either cultivation setting (PBR or pond) or light source types. It can be transferred to any types of cultivation settings as long as we collect enough training samples (e.g., LDP for LDPM training, and growth data for GRM training). Actually, such versatility is a significant advantages of machine learning-based models over other types of models. We have discussed it by adding a sentence ‘The methodology for LDP prediction proposed in this study could be transferred to any existing algal cultivation systems, such as indoor/outdoor PBRs or pond systems.’ in line 111-113.

Furthermore, we have proven this statement by performing additional experiments to demonstrate the application of machine learning-based models for LDP and growth prediction in pond systems with changing light. Our results suggested that machine learning models can accurately predict LDPM and growth rates in the pond system as shown in figure 6. we have described the results by adding sentences ‘We first adapted the machine learning models (LDPM and GRM) for a pond system to guide the cultivation design. Both models showed high prediction accuracy. The LDPM achieved an overall R-squared score of 0.986 (Figure 6B) and pixel-by-pixel analysis suggested the LDP prediction was reasonably good at all pixels, with a minimal R-squared score of 0.943 (Figure 6C). The GRM prediction also achieved an R-squared score of 0.980 (Figure 6E).’ in line 317-322.

Figure 6 (B, C, E, F.) B and C shows the prediction evaluation of the adapted LDPM for pond system. The LDPM achieved an overall R-squared score of 0.986 and pixel-by-pixel analysis suggested that prediction at all pixels were reasonably good, with a minimal R-squared score of 0.943. C, the biomass production with the outdoor pond system achieved a record biomass productivity. E, machine learning-based growth simulation suggested that setting the initial cell concentration at around 0.4 g/L achieves optimal biomass productivity, under the growth condition mimicking Texas Summer, which in accord with the measured results.

Comment-2a: The model suggested that light is a key factor for cell growth, which was not fully demonstrated or described.

Response:

We appreciate the comments very much. Because we tightly controlled the temperature and nutrients supply during the tests, the light availability would be the major factor limiting the cell growth. Light as a key limiting factor for growth under sufficient nutrient has been well documented by previous studies^{4,5} and the phenomena was also well supported by the machine learning outputs (Figure S3). In Figure S3, we found light patterns is strongly correlated to the cell growth rates. We recognize that the results were not fully described in the previous version. We have revised this part with more detailed description and highlighted the correlation to previous data. The part now read: ‘We found that the shading effect increased sharply when cells grow to a high concentration (Figure S2), similar to previous studies^{1,6}. Cyanobacterial growth rates peaked when dark areas, defined as pixels with GSVs less than 25.5 (10% of the maximal value), reached $43.0 \pm 8.9\%$ at all tested light conditions. The growth rate dropped drastically when dark areas reached a plateau around 65% (Figure S3). Specifically, when dark areas reached 43%, cell growth began to be inhibited by mutual shading. Such inhibition intensified after dark areas reached 65%. The strong correlation between light pattern and growth rates suggests that light availability is the primary factor determining cyanobacterial growth rates when nutrients are sufficient and temperature is controlled. The results are consistent with previous findings that light availability defines the growth potential for cyanobacteria given abundant nutrients. More importantly, this quantitative understanding allowed us to develop a second machine learning model to predict growth rate based on LDPs.’ in line 119-131.

Comment-2b: The authors simply explained the Fig. 2D-F that they were able to predict the productivity based on initial OD and light. I am not sure that how this model is superior to a Monod type equation in general at a fixed light source. This point, comprehensive optimization must be followed to support the predictive model, which lacks in this study.

Response:

We appreciate the comments about the strength of the machine learning models. First of all, we have thoroughly discussed the advantages of the machine learning models in responses to comment 1a. Moreover, Monod type equation requires system-wide understandings of kinetic behaviors and usually simplified with certain assumptions to reduce complexity.⁷ Such simplification would significantly impact the prediction performance when applied to algal prediction,⁸ especially when changing light is included. In particular, the ML-model has higher accuracy, larger OD/cell concentration range, multi-dimensional capacity, and superior flexibility with different light conditions, cultivation systems, and factors. The superior capacity of machine learning models allowed us to integrate LDP model and growth model to simulate cyanobacteria growth at a given initial OD, in either fixed or changing light conditions, which will be challenging for Monod type equation. Such a novel capacity solved a major challenge in developing a viable and efficient semi-continuous cultivation system, as such a system requires to determine how much cell to harvest at each cycle to achieve an optimal initial inoculum size for next round growth to reach maximum speed. The Figure 2F actually shows that the machine learning models could predict the optimal initial OD under changing light condition. In response to the reviewer's suggestion, we have discussed the advantage of machine learning models over mathematical models as follows. 'Empowered by machine learning models, we were able to simulate cyanobacterial growth under different constant light conditions by combining the LDPM and GRM (Figure 2A). As shown in Figure S4 A-F, the simulated growth was very close to measured growth at all tested conditions, with a lowest R-squared value of 0.996. We also tested if the machine learning models could simulate cyanobacterial growth under changing light conditions. As shown in Figure S4G, growth predictions under changing light achieved an R-squared score of 0.978 compared to measured results, validating the accuracy of the model. Overall, the results demonstrated that machine learning models could accurately simulate cyanobacterial cell growth at both constant and changing light conditions. The machine learning model is thus more versatile compared to traditional mathematical models (e.g., models based on the Monod equation) and does not require prior knowledge of growth characteristics. Moreover, the machine learning-based growth simulation is highly flexible and could expand to integrate other growth impacting factors such as temperature and nutrients. Such integration might be too complicated for traditional mathematical models, especially under changing light.' in line 146-161.

Furthermore, we appreciate the reviewer's suggestions on model optimization and have carried out thorough model optimization to improve the prediction accuracy of the LDPM by adding regulation and to improve the GRM by replacing the support vector regression algorithm with random forest algorithm. The optimizations have

improved the prediction performance of the LDPM (R-squared score increased from 0.993 to 0.995) and GRM (R-squared score increased from 0.971 to 0.990), and enabled accurate growth simulation under both fixed light and changing light conditions (Figure S4 A-G). The Figure S4 below highlighted the model accuracy and the changing light prediction capacity (Figure S4G).

Figure S4 A-G. Comparison of measured growth with growth simulation under different light conditions. A, $107 \mu\text{mol m}^{-2} \text{s}^{-1}$, $R^2 = 0.996$. B, $178 \mu\text{mol m}^{-2} \text{s}^{-1}$, $R^2 = 0.999$. C, $267 \mu\text{mol m}^{-2} \text{s}^{-1}$, $R^2 = 0.996$. D, $357 \mu\text{mol m}^{-2} \text{s}^{-1}$, $R^2 = 0.998$. E, $570 \mu\text{mol m}^{-2} \text{s}^{-1}$, $R^2 = 0.996$. F, $714 \mu\text{mol m}^{-2} \text{s}^{-1}$, $R^2 = 0.997$. G, changing light, $R^2 = 0.978$.

Comment-3a: There was limited information about the SAC system given in this manuscript. Thus, it cannot be evaluated for the performance and feasible operation. This also limits understanding of how the ML assists the SAC.

Response:

We appreciate the comments, guiding us to improve clarity in our presentation. According to the comments, we added a figure (Figure 5A) to illustrate the concept of the SAC system and the integration of the machine learning-informed SAC with the ABS. We have also revised the Material and Method with more detailed information. The description of the SAC PBR system in line 425-433 is now reads: 'A customized PBR (based on a 1-L Roux bottle) containing 500 ml of media was used for cultivation, with 5% (vol/vol) CO₂ bubbling from a stainless-steel aeration stone at a speed of 0.8 L/min. 10 ml of 50× stock media was fed every 24 hours for fed-batch cultivation. For SAC, initial cell concentration was adjusted to OD₇₃₀ of ~2.3 every 24 hours followed by media feeding. The initial OD was selected based on the machine learning model outcome of optimal starting OD. The growth temperature of batch cultivation, fed-batch, and SAC was maintained at 37°C and artificial light at $574 \mu\text{mol m}^{-2} \text{s}^{-1}$ was applied on two opposite sides of the PBR.'

Figure 5A. The concept of the SAC system and the integration of the machine learning-informed SAC with the ABS.

Furthermore, we also expanded the SAC to pond system according to the request of the reviewer. We have also included detailed information about the pond SAC system in line 433-445. The information reads: ‘A customized pond system was used for scaling-up of the SAC, shown in Figure 6A. The circular pond system contained a 6-inch-wide raceway and an impeller was used to keep the cyanobacterial cells agitated. 30 liters of cyanobacteria (20 cm in height) were cultivated in the pond system with 5% CO₂ (vol/vol) bubbling via gas dispersion stones. The growth temperature was maintained at 40°C with a water heater. Cell growth and light conditions were monitored with a turbidity meter (EXcell231, EXNER) and a light sensor (LS-BTA, Vernier), respectively. In the condition mimicking Texas summer, the pond system was placed in a growth chamber and the light program was set to 400 μmol m⁻² s⁻¹ for 1h, 800 μmol m⁻² s⁻¹ for 1h, 1300 μmol m⁻² s⁻¹ for 1h, 1500 μmol m⁻² s⁻¹ for 10h, 1300 μmol m⁻² s⁻¹ for 1h, 800 μmol m⁻² s⁻¹ for 1h, and 400 μmol m⁻² s⁻¹ for 1h (all light intensities were measured from the pond surface). In both outdoor and mimicking outdoor condition, 250 ml water was added to the pond system every 2 hours to counter evaporation.’

Figure 6A. Demonstration of the pond system setup.

Comment-3b: Besides, no applications were performed using the SAC., except the limonene production below. An independent application must be followed to demonstrate the usefulness.

Response:

We appreciate the comments. The main innovation in this research is using machine learning to guide the SAC design and using the ABS to reduce harvesting costs, to enable economic production of fuel precursors. The combining of the two aspects have created significant novelty. With the SAC, we have achieved record biomass productivities from both indoor PBR and outdoor pond systems. With the high productivity and low harvesting cost, we have brought algal biomass production cost to below \$300. The economics opens enormous opportunity for algal biomass

applications. Previous studies have highlighted several potential applications of algal biomass, including bioconversion to fuels, animal feed, and thermochemical conversion.⁹⁻¹¹ Regardless of the applications, the high productivity of algal biomass and low production cost are the key to downstream applications. The manuscript has solved this bottleneck. The biomass contents 36.5% of carbohydrates and thus can be used as feedstocks for ethanol fuel production. Moreover, limonene produced from the PBR could also be used as ‘drop-in’ jet fuel. Furthermore, the remaining biomass can serve as nutritious animal feed due to the high protein content (42.8%).^{11,12} In response to the reviewer’s request, we have expanded the discussion on the applications ‘At the same time, the SAC process would generate biomass that is significantly less costly to convert to ethanol than the current most common feedstock (corn), as it would eliminate the need for costly milling and other pre-treatment prior to fermentation.’ in line 384-387 and ‘Furthermore, the lower cost of algal biomass enables economically competitive applications in broader industries, including algal biofuel, animal feed, food additives, and various specialty products.’ in line 417-419.

Comment-3c: In Fig. 2G, biomass was compared to the FB as a control, in terms of productivity (g/L/d). No sufficient data are given (i.e. growth, CO₂ uptake or conc., Chl a, etc over time).

Response:

We appreciate the comments very much. As suggestion, we have updated the Figure 2G and provided the corresponding growth data in Figure S4H. The results complemented with the Figure 2G and showed that effectiveness of the SAC in maintaining biomass productivity. The Figure S4H has been quoted in the sentence ‘The growth of cyanobacteria in fed-batch and SAC is shown in Figure S4H’ in line 198-199.

FigureS4H. The growth of cyanobacteria in fed-batch and SAC.

Comment-4a: Authors claimed that engineered strain exhibits increased hydrophobicity by producing limonene. Then, the droplets were analyzed via several tools including TEM, SRS and, BATH. Previously, limonene was volatile and their collections from the medium have been studied. However, this study showed that hydrophobic products were found on the surface, acting as an aggregating agent. Although various analyses have demonstrated the hydrophobicity of limonene-producing cells, the SAC culture was not fully described for this reviewer to evaluate the novelty and impact of the system.

Response:

We appreciate the comments. Our TEM, SRS and, BATH results are actually not conflicted with previous studies. Instead, they are complementing with previous discoveries. Although previous studies showed that limonene is volatile and can be collected from either medium or headspace, it is still not clear how limonene molecule

is volatilized from cyanobacteria cells. Our TEM and SRS results suggested that limonene was first secreted from the cell as droplets and then possibly, volatilized from the media due to immiscibility. The presence of limonene on cell surface could increase cell hydrophobicity, which had been well supported by the BATH assay. The aggregation tests (Figure 3A) and the sedimentation measurements (Figure 4 and supplementary video File S2) also verified the limonene-mediated aggregation, as the presence of a limonene synthase is the only genetic difference between L524 and WT for UTEX2973 strain.

Two important factors contributed to the ABS effects in UTEX 2973: 1) the much higher limonene productivity in UTEX 2973 as compared to PCC7942 (1.4 mg/l/OD/day Vs. 0.8 mg/l/OD/day), and 2) the unique 'smooth' cell surface structure of the UTEX 2973. Due to the mutation of pilN protein, the UTEX 2973 does not have pilus on its surface. The comparison between cell surface of UTEX 2973 and PCC7942 clearly showed that UTEX 2973 has flatten surface without significant pilus formation. Such flatten surface together with the hydrophobic limonene droplets could allowed the hydrophobic interaction between the cells to aggregate together due to the hydrophobic interaction in a water environment. We have expanded the discuss of ABS and hydrophobic interaction in the revised manuscript by adding sentences 'Moreover, the unique 'smooth' cell surface of UTEX 2973 might also promote aggregation in combination with the hydrophobic interaction of limonene production. Unlike other cyanobacteria, pili rarely form on the UTEX 2973 cell surface (Figure S6), presumably due to the early termination of the pilN protein¹³. The flatter cell surface of UTEX 2973 allows limonene droplets among different cells to interact with one another more easily compared to strains like PCC 7942 (Figure S6). Together, limonene production and the smooth cell surface might have enabled the engineered cells to aggregate due to hydrophobic interaction in a water environment.' In line 254-262.

As for the SAC, we have provided more detailed information including a concept figure in the revised manuscript, as described in Comment 3a.

Comment-5: Authors claimed that the SAC platform is economically feasible. This evaluation was based on the indoor culture with limited scales. Thus, it is not proper to evaluate the TEA.

Response:

We appreciate the comments. As suggested, we have carried out outdoor cultivation with a pond system, and redone the TEA analysis with the updated data from the open pond system. The updated TEA analysis suggested that the SAC could deliver a MBSP approximately \$281 per ton ash free dry weight. The result is highly consistent with our previous TEA analysis of below \$300 per ton biomass and indicates the economic feasibility. The parts now reads 'Based on the NREL study, the yearly average of biomass productivity is estimated to be the productivities achieved in the Spring (MAR, APR, MAY) and Fall (SEP, OCT, NOV).¹⁴ Following that approach, we estimated the yearly average of biomass productivity for the open pond system to be 43.3 g/m²/d in the outdoor study and 48.1 g/m²/d (83.3% of summer productivity) in the indoor mimicking trial. The ash content of the cyanobacterial biomass was measured to be 5.5%. At these conditions, the NREL model projects a MBSP of approximately \$281 per ton based on the outdoor trial yield (Figure S9)' In Line 350-357.

Comments from Reviewer 2

The manuscript entitled “Machine Learning-Informed and Synthetic Biology-Enabled Semi-Continuous Algal Cultivation (SAC) to Unleash Renewable Fuel Productivity” applied ML methods to predict light spatial distribution and the microalgal growth rates. Also, they designed a low-cost biomass harvesting process, which is one of the main challenges for microalgal production. In my opinion, this study is within the scope of Nature Communications. It is a very interesting paper, and I recommend its publication after minor changes. Please see my comments below.

Response:

We appreciate the reviewer’s comments about study being ‘very interesting’ and ‘within the scope of Nature Communications’. We also appreciate that the reviewer acknowledged that we addressed ‘one of the main challenges of microalgal production’ and recommend the publication with minor changes.

Comment-1a: L81: How were greyscale values calculated?

Response:

We appreciate the comments about the grayscale value calculation. To extract the grayscale values, we first transformed the LDP images (RGB format, mostly green) to grayscale images and extract the grayscale value with the CV2 module in Python. We have made the modification to clarify the process and described it in the Supplementary Methods line 35-37. The sentences read: ‘The compressed images were used to represent the light distribution pattern inside the photobioreactor with grayscale values (GSVs) representing light intensities. The GSVs were extracted from the grayscale images with the CV2 module in Python.’

Comment-1b: Was the light intensity measured at the surface of PBR (and in the two dimensions)?

Response:

We appreciate the comments to allow us clarify the methodology. Yes, the light intensity was measured at the surface of the PBR. It was measured with a light sensor and is not two dimensional. We have updated the Supplementary Methods section by adding a sentence ‘The illuminance was monitored by a sensor on the surface of the photobioreactor and converted to photosynthetic photon flux density (PPFD) with a coefficient at of 56.’ in line 26-28.

Comment-2: L93: How were these percentages calculated? R-squared values are calculated for a set of values. Do you divide the data to determine different R-squared values?

Response:

We appreciate the comments percentage and R-squared score calculation. The question reminded us to improve clarity of the presentation. For machine learning modeling, one often breaks down the data to training set and testing set. Once the training is completed, the predicted value will be generated for testing set, while the testing set also has the actual experimental value. This allows to investigate the correlation between the predicted and experimental value, which generated the R-square that we presented. Such R-square describes the accuracy of the modeling. In order to train the LDPM, we have collected 138 LDP images as training samples. We reserved ~10% of LDPs as testing set. Since LDPs are 40 × 18 (pixel) images, we trained 720 machine learning models for pixel-by-pixel LDP prediction. We used the

testing set to test the performance of the model predictions as aforementioned. The R-squared score for each pixel is calculated by comparing the set of predicted values with actual values at that pixel. The results of R-squared score for each pixel is shown in Figure 1B. As for the calculation of percentages, we counted the number of pixels with a score large than 0.90 (or between 0.79 and 0.85), and divided the number by 720. We have updated the Supplementary Methods with detailed information by adding sentences ‘Prediction accuracy was determined by overall evaluation and by pixel-by-pixel evaluation. The overall evaluation calculated an R-squared value by comparing all predicted GSVs with measured GSVs in the testing data set to assess the overall prediction accuracy of the model. Pixel-by-pixel evaluation calculated the R-squared value at each pixel to assess the prediction accuracy at different positions on LDPs. Accuracy percentages were calculated by counting pixels with an R-squared score larger than 0.90 (or between 0.79 and 0.85), and dividing by 720. The R-squared evaluation was performed with the metrics module on scikit-learn.’ In line 63-70. We also described the selection of 10% of total dataset as testing data set in line 60 of supplementary data.

Comment-3: L112: How was the correspondence between LDP and specific growth rates attributed?

Response:

We appreciate the comments. The correspondence is based on the light availability and growth rate at a certain time point. Basically, at each time point, we know the cyanobacterial cell concentration and the light intensity, so we can predict the LDP using the LDPM. Meanwhile, we can also calculate the growth rate at the same time point from growth curve. We set the predicted LDPs as features and the calculated growth rates as labels to training the GRM for growth rate prediction. The growth simulation basically iterates the process of predicting LDP, growth rate, cell concentration, as shown in Figure 2A. We have updated the sentence with specified correspondence. The sentence now reads: ‘Vectors extracted from LDPs and their corresponding growth rates (based on the same time points) were set as features and labels in the training, respectively.’ in line 133-135.

Comment-4: L175: And the density of the lipids? What is the mass fraction of lipids on biomass?

Response:

We appreciate the comments about the lipid. The density of the lipids is 860 kg/m³. And the mass fraction of the lipids on biomass is 11.2%. The strength of UTEX 2973 is the high glycogen content for potential ethanol fermentation and the high protein content for animal feed. The lipid content is relatively low. Furthermore, the high protein and carbohydrate content also allows ABS. We have added the information in the revised manuscript in line 212-216. The sentences now read ‘UTEX 2973 cells contain around 42.8% protein, 36.5% carbohydrates, and only 11.2 % lipid. Due to the high carbohydrate content (average density around 1500kg/m³), high protein content (average density around 1300kg/m³), and low lipid content (average density around 860 kg/m³) of UTEX 2973 cells, they should be dense enough for sedimentation in water (~1000kg/m³).’.

Comments from Reviewer 3

This study investigated using machine learning models to predict light availability and algae cells growth rates, which enables the design of Semi-continuous Algal Cultivation (SAC) to achieve high algae productivity. In addition, to realize low cost algae harvesting under SAC mode, aggregation-based sedimentation (ABS) was developed via engineering a fast-growing strain to generate limonene to enables efficient cell aggregation and sedimentation. The combination of SAC and ABS via engineering strains to generate limonene is very interesting and innovative. This method can greatly reduce the algae cultivation cost via high productivity and low cost harvesting. The research methods used in the study well supported major results and conclusions, except for the cost analysis part. Therefore, the manuscript needs modifications before being considered for publication.

Response:

We appreciate the reviewer's comments about that 'The combination of SAC and ABS via engineering strains to generate limonene is very interesting and innovative' and 'The research methods used in the study well supported major results and conclusions.' We have revised the manuscript and redone the TEA with NREL model with updated outdoor growth data from a pond system to generate the reliable cost analysis.

Comment-1: Line 158 to 159, as claimed by the authors, "we set up the SAC under double light sources", this SAC mode based on the machine-learning model developed in this study primarily can only be used for photobioreactor (PBR) with double light sources. This limited the application of this machine-learning model to other types of algae cultivation systems, such as open pond and single-slight source PBR.

Response:

We appreciate the comments about adaptability of the machine learning models very much. In fact, the machine learning models are very flexible, and can theoretically be applied into any cultivation systems. We have discussed the versatility of the machine learning models in the revised manuscript by adding a sentence 'The methodology for LDP prediction proposed in this study could be transferred to any existing algal cultivation systems, such as indoor/outdoor PBRs or pond systems.' in line 111-113.

Furthermore, as a demonstration of its versatility, we have adapted machine learning models for PBR with one light source (Figure 1 and 2, with both constant light and changing light), PBR with double light sources (Figure S5), and indoor/outdoor open pond systems (Figure 6). The results suggested that the machine learning-based predictions is a versatile tool for guiding agal cultivation design. In particular, we have carried out extensive experiments to demonstrate the applicability of the machine learning models in pond system, which allows better TEA evaluation. The description of pond system modeling and SAC experiments are as follows. 'We further validated the potential of SAC with a 30-liter raceway pond system. We first adapted the machine learning models (LDPM and GRM) for a pond system to guide the cultivation design. Both models showed high prediction accuracy. The LDPM achieved an overall R-squared score of 0.986 (Figure 6B) and pixel-by-pixel analysis suggested the LDP prediction was reasonably good at all pixels, with a minimal R-squared score of 0.943 (Figure 6C). The GRM prediction also achieved an R-squared score of 0.980 (Figure 6E). Like the PBR system, we employed the machine learning models to predict optimal initial cell concentrations for the pond SAC system. The growth simulation suggested that setting initial cell concentration to around 0.4 g/L

delivers the highest biomass productivity under the growth condition mimicking Texas summer (Figure 6F). Based on the prediction, the experimental results showed that SAC achieved the highest biomass productivity at 58.1 g/m²/d (Figure 6F).’ in line 317-328.

Comment-2: Line 165 to 168: First, this is not ‘the first reported machine-learning-informed algal cultivation design.’ The authors should do more literature searches. Although very limited, at least one report has been published about using machine-learning models to guide algal cultivation design based on light distribution. Please modify this claim.

Response:

We appreciate the reviewer pointing out the wrong claim. As suggested, we have deleted that sentence.

Comment-3: Also together with comment 1, the application of this machine-learning models with double light sources is limited considering various algae cultivation methods. Please discuss the possible improvement that can be done for the machine-learning models.

Response:

We appreciate the comments. As discussed in comment 1, we have demonstrated that the machine learning model is very versatile and can be used for predictions with various cultivation setting. However, we do agree that the machine learning model can be further expanded to more complicated conditions. We hereby discussed possible integration of the machine learning-based growth simulation by adding a sentence ‘Moreover, the machine learning-based growth simulation is highly flexible and could expand to integrate other growth impacting factors such as temperature and nutrients. Such integration might be too complicated for traditional mathematical models, especially under changing light.’ in line 158-161. Furthermore, we also expanded the discussion of machine learning model adaptability and improvement at the end. ‘The research has led to several breakthroughs that could have a profound impact on biomanufacturing, algal bioproduction, and renewable fuels and products. First, the study is one of the first to use Artificial Intelligence techniques to guide algal cultivation design. In particular, the research provided the first-ever quantitative insights into how light intensities and cell density shape LDPs and how LDPs, in turn, impact cyanobacterial growth rates. The integration of LDPM and GRM enables reliable simulation of growth curves based on initial OD and light intensity. This new knowledge inspired us to develop SAC and precisely defined the optimal initial OD to achieve maximized growth. The high accuracy, broad prediction range, and superior capacity to handle complexity of machine learning models produced broader adaptability in constant or changing light and in indoor/outdoor PBRs or pond systems. The principle and design of the study can be broadly applied to industrial microbiology and biomanufacturing. The machine learning models themselves can be broadly adapted to different set-ups to guide algal cultivation management and design. The models can be further optimized to integrate nutrient, temperature and other factors to achieve even broader adaptability.’ in line 389-403.

Comment-4: Line 175 to 178: This sentence is awkward and confusing. High enough density is typically needed for sedimentation no matter what carbohydrates and proteins contents of the algae. Please rewrite this sentence and clearly describe the

carbohydrates/proteins contents impacts on cell densities.

Response:

*We appreciate the reviewer to guide us to improve the readability. Actually the ‘density’ in the sentence does not mean concentration. It means how ‘heavy’ the cells are. Due to the high content of carbohydrates and protein, both of which are ‘heavier’ than water, the cyanobacterial cell thus should be ‘heavy’ enough for efficient sedimentation. We have revised the sentence to make it clearer. The sentence now reads ‘**UTEX 2973 cells contain around 42.8% protein, 36.5% carbohydrates, and only 11.2 % lipid. Due to the high carbohydrate content (average density around 1500kg/m³), high protein content (average density around 1300kg/m³), and low lipid content (average density around 860 kg/m³) of UTEX 2973 cells, they should be dense enough for sedimentation in water (~1000kg/m³).**’ In line 212-216.*

Comment-5: Line 227: This is a big achievement. If the ABS output cells concentration can reach 139.2 g/L, it will greatly reduce the harvesting cost by lower dewatering cost and energy needs.

Response:

We appreciate the reviewer recognizing that the high solid output from the ABS is a ‘big achievement’ and ‘will greatly reduce the harvesting cost’.

Comment-6: Line 229 to 231: this conclusion might need a longer testing time to be verified, such as 10 days or 1 month. As air or CO₂ bubbling is not always on, there might be some aggregation happened during the growth phase compared to the wild type. If so, it will affect the light adsorption of the algae cells.

Response:

We appreciate the comments very much. We have replaced the figure with updated 7-day growth results. In fact, the air/CO₂ bubbling is always on during the cultivation, and we did not find significant growth inhibition caused by aggregation during the 7-day cultivation (Figure S7). The reason we didn’t provide longer data (>10 days) is that we found cell growth was seriously inhibited by mutual shading after day 3, and should be harvested in the SAC system to maintain its high productivity.

Figure S7. Comparison of growth between UTEX 2973 WT and L524. No significant growth differences were found between WT and L524 in the given growth conditions.

Comment-7: Line 252: Is the productivity dry basis or dry ash free basis? If dry basis, please provide dry ash free basis or ash % information.

Response:

*We appreciate the comments. The productivity is dry weight basis, and we have provided the ash % information in line 355 by adding a sentence ‘**The ash content of the cyanobacterial biomass was measured to be 5.5%.**’ In the updated manuscript.*

Comment-8: Line 281 to 282: The “40 g/m²/d” productivity is different from my calculation results based on the “0.1 g/L/h” in your study and information in ref 45. Please explain how the 0.1 g/L/h is translated into 40 g/m²/d.

Response:

We appreciate the comments about the productivity calculation. In fact, the productivity could be much higher than 40 g/m²/d based on the 0.1 g/L/h, depending on the photobioreactor configuration. Based on the reviewer’s suggestions, we have performed additional experiment with an outdoor pond system. The pond system provided a very direct yield calculation and achieved an average biomass productivity of 43.3 g/m²/d with the SAC system (Figure 6D). We have discussed both the indoor and outdoor experiments for summer and fall, respectively. It should be noted that the indoor and outdoor data are very consistent when the indoor summer data were converted to the yield in the fall. We have described the indoor and outdoor pond experiments as follows in line 326-343. ‘Based on the prediction, the experimental results showed that SAC achieved the highest biomass productivity at 58.1 g/m²/d (Figure 6F). We noticed slight differences between the predicted biomass productivity and measured productivity when initial cell concentrations were around 0.4 g/L (Figure 6F). The deviation might result from the presence of noise in the training data, and/or overfitting in the models. Future optimization such as removing noise, adding regulations, and expanding training data could further enhance the model performance. Overall, our results demonstrated the application of machine learning models in a pond SAC system. The success of application in both PBR and pond systems indicates that machine learning-based prediction can be a generalized method for guiding algal cultivation management and design in various systems.

Inspired by the high productivity from the indoor pond system, we further tested biomass productivity of the pond SAC in real outdoor conditions. The outdoor tests were carried out in late September 2021 in College Station, Texas, with both ‘partially sunny’ and ‘mostly sunny’ weather. These conditions represent a typical fall growth condition. The outdoor cultivation achieved an average biomass productivity of 43.3 g/m²/d (Figure 6D), which, to our knowledge, represents a new record of algal biomass productivity in outdoor open pond systems.’

Furthermore, we have discussed the consistency of the pond data in line 350-355. ‘Based on the NREL study, the yearly average of biomass productivity is estimated to be the productivities achieved in the Spring (MAR, APR, MAY) and Fall (SEP, OCT, NOV).¹⁴ Following that approach, we estimated the yearly average of biomass productivity for the open pond system to be 43.3 g/m²/d in the outdoor study and 48.1 g/m²/d (83.3% of summer productivity) in the indoor mimicking trial.’

Comment-9: Line 285: Please define or provide the major cost parameters for this “break-even MBSP” estimation. For example, what is the internal rate of return for this “break-even” MBSP, 0%? In addition, to translate to a horizontal tube PBR system, what is the productivity for the MBSP estimation?

Response:

We appreciate the comments, which helped us to improve the preciseness of the TEA part. First, in the revised manuscript, we note that NREL employed a 10% internal rate of return in their estimation in line 348-350, saying

‘Both studies focused on estimating the break-even minimum biomass selling

price. (MBSP), given an internal rate of return on capital of 10%.’

Second, regarding to MBSP, we have followed this reviewer’s and reviewer 1’s suggestion to expand the Machine Learning-informed SAC to the pond system. In the pond system, we actually obtained out-door productivity experimentally, which is 43.3 g/m²/d. The experimental data, instead of translated data, has provided a more solid estimation of MBSP. We have included the discussion as follows.

‘Based on the NREL study, the yearly average of biomass productivity is estimated to be the productivities achieved in the Spring (MAR, APR, MAY) and Fall (SEP, OCT, NOV).¹⁴ Following that approach, we estimated the yearly average of biomass productivity for the open pond system to be 43.3 g/m²/d in the outdoor study and 48.1 g/m²/d (83.3% of summer productivity) in the indoor mimicking trial, respectively. The ash content of the cyanobacterial biomass was measured to be 5.5%. At these conditions, the NREL model projects a MBSP of approximately \$281 per ton based the outdoor trial yield (Figure S9). By comparison, 2019 state-of-the-art open pond algal cultivation had an MBSP of approximately \$1,227 per ton.’ in line 350-358.

Comment-10: Line 299 to 300: “quite possibly to below \$300 per ton AFDW.” This conclusion cannot be clearly obtained from the data or inputs provided by the authors. At least, some major cost assumptions directly related to this conclusion should be clearly provided and described. For example, what is the PBR system assumed for this MBSP? What is the productivity? What are the reference values (productivity and MBSP) used as a starting point or basis for the result of \$300/ton AFDW? Are there other major cost assumptions used here? No robust cost estimates is requested, but the authors should clearly demonstrate the cost analysis basis for the \$300/ton AFDW MBSP result.

Response:

We agree that the previous version of the manuscript was unclear in this regard, and thank the reviewer for pointing this out. As mentioned in comment 10, we have provided major cost assumptions in the latest TEA Material and Methods along with Supplementary Figure 9. We have also updated the cost analysis part in the results section. The system assumed is open pond based on the reviewer’s previous request. The productivity and reference values were all provided, enabling robust cost estimates with a specific value. Furthermore, we discussed the factors that can further reduce the MBSP.

The part now reads ‘Based on the NREL study, the yearly average of biomass productivity is estimated to be the productivities achieved in the Spring (MAR, APR, MAY) and Fall (SEP, OCT, NOV).¹⁴ Following that approach, we estimated the yearly average of biomass productivity for the open pond system to be 43.3 g/m²/d in the outdoor study and 48.1 g/m²/d (83.3% of summer productivity) in the indoor mimicking trial, respectively. The ash content of the cyanobacterial biomass was measured to be 5.5%. At these conditions, the NREL model projects a MBSP of approximately \$281 per ton based the outdoor trial yield (Figure S9). By comparison, 2019 state-of-the-art open pond algal cultivation had an MBSP of approximately \$1,227 per ton. The categorical cost distribution is shown in Figure S9.

Furthermore, the limonene produced by L524 has a current market value of about \$5/kg. At this price, the SAC system proposed here would generate approximately \$10.08 of additional revenue in limonene sales per ton of biomass produced. Such reductions in MBSP can be readily achieved in PBR systems. Although limonene collection from open pond systems may not be cost effective at

current productivity levels, limonene-mediated ABS nonetheless significantly reduces harvesting costs.

Beyond significant improvements in biomass production, the implementation of ABS in SAC would also markedly reduce operating costs. ABS (0.1 kWh m^{-3}) could save up to 93% on energy costs compared to traditional harvesting methods (e.g., disc stack centrifugation (1.4 kWh m^{-3})),³ while maintaining high efficiency and recovery rates. As the dewatering process accounts for \$24.4 per ton of biomass in the current model (Figure S9), the simplified harvest by ABS would further significantly reduce the MBSP (however, we have not adjusted the \$281 per ton MBSP generated by the NREL model to reflect such reductions).’ in line 350-373.

The Figure S9 is as shown below with defined breakdown of cost.

Figure S9, cost breakdown shows that the dewatering process accounts for \$24.4 per ton of biomass in the current model.

Comment-11: Figure 1 and 2 have some repetitive information. Figure 1 A and figure 2 A should be combined into a single figure. Figure 1 C, D and E seems redundant, which are suggested to use only one figure to represent the R-squared scores for pixels prediction.

Response:

We appreciate the comments about the repetitive information in figures. The comment actually helped us to improve the presentation. Although there are a little bit overlapping between Figure 1A (concept figure of LDPM) and Figure 2B (previous Figure 2A, concept figure of GRM), the two figures represent two separate machine learning models. The first one is for LDP prediction and the second one is GRM prediction. The design of the two models is quite different. We did not to combine the two models because it could bring up some confusions, especially the GRM is based on the LDPM. Nevertheless, we do realize the possible confusion caused by two machine-learning models. Therefore, we moved Figure 2A (previously Figure 2B) up and expanded it with final output of optimal OD for SAC design. Figure 2A shows the integration of the two machine learning models (LDPM and GRM) and the output of model integration to guide SAC design. The revised Figure 2 is as shown below to improve the logic flow.

Figure 2. Growth rate prediction, growth simulation and the semi-continuous algal cultivation.

We agree that Figure 1B (previous Figure 1C) is somehow redundant with the Figure S1 (previous Figure 1D), so we have moved the Figure 1D to supplementary data. The Figure 1D (previous Figure 1E) is actually different from the Figure 1B (previous Figure 1C). The Figure 1B shows the prediction accuracy at different positions on predicted LDPs, and it tells us the predictions at which pixels are more or less accurate. The Figure 1D shows the overall prediction accuracy, and it tells us how are the predicted LDPs similar to the actual LDPs. Thus, we kept the two figures in the revised manuscript. The revised Figure 1 based on Reviewer’s comment is as shown here.

Figure 1. Data processing and machine learning.

Comment-12: Line 285: “of” should be “to”.

Response:

We appreciate the comments. we have deleted the sentence in the revised manuscript.

References

- 1 Kumar, K., Sirasale, A. & Das, D. Use of image analysis tool for the development of light distribution pattern inside the photobioreactor for the algal cultivation. *Bioresource Technol* **143**, 88-95, doi:10.1016/j.biortech.2013.05.117 (2013).
- 2 Lee, C.-G. Calculation of light penetration depth in photobioreactors. *Biotechnology and Bioprocess Engineering* **4**, 78–81, doi:<https://doi.org/10.1007/BF02931920> (1999).
- 3 Suh, I. S. & Lee, S. B. A light distribution model for an internally radiating photobioreactor. *Biotechnol Bioeng* **82**, 180-189, doi:10.1002/bit.10558 (2003).
- 4 Simionato, D., Basso, S., Giacometti, G. M. & Morosinotto, T. Optimization of light use efficiency for biofuel production in algae. *Biophys Chem* **182**, 71-78, doi:10.1016/j.bpc.2013.06.017 (2013).
- 5 Clark, R. L. *et al.* Light-optimized growth of cyanobacterial cultures: Growth phases and productivity of biomass and secreted molecules in light-limited batch growth. *Metabolic Engineering* **47**, 230-242, doi:10.1016/j.ymben.2018.03.017 (2018).
- 6 Yen, H. W. & Chiang, W. C. Effects of mutual shading, pressurization and oxygen partial pressure on the autotrophical cultivation of *Scenedesmus obliquus*. *J Taiwan Inst Chem E* **43**, 820-824, doi:10.1016/j.jtice.2012.06.002 (2012).
- 7 Liu, S. in *Kinetics, Sustainability, and Reactor Design* (ed Shijie Liu) Ch. 11, 629-697 (Elsevier, 2017).
- 8 Hossain, S. M. Z. *et al.* in *Renewable Energy and Sustainable Buildings: Selected Papers from the World Renewable Energy Congress WREC 2018* (ed Ali Sayigh) 517-528 (Springer International Publishing, 2020).
- 9 Aikawa, S. *et al.* Direct conversion of *Spirulina* to ethanol without pretreatment or enzymatic hydrolysis processes. *Energy Environ Sci* **6**, 1844-1849, doi:10.1039/c3ee40305j (2013).
- 10 Aikawa, S. *et al.* Direct and highly productive conversion of cyanobacteria *Arthrospira platensis* to ethanol with CaCl₂ addition. *Biotechnol Biofuels* **11**, 50, doi:10.1186/s13068-018-1050-y (2018).
- 11 Becker, E. W. Micro-algae as a source of protein. *Biotechnology Advances* **25**, 207-210, doi:10.1016/j.biotechadv.2006.11.002 (2007).
- 12 Chojnacka, K., Wieczorek, P. P., Schroeder, G. & Michalak, I. Algae Biomass: Characteristics and Applications Towards Algae-based Products Preface. *Devel Appl Phycol* **8**, V-Vi, doi:Book_Doi 10.1007/978-3-319-74703-3 (2018).
- 13 Li, S. B., Sun, T., Xu, C. X., Chen, L. & Zhang, W. W. Development and optimization of genetic toolboxes for a fast-growing cyanobacterium *Synechococcus elongatus* UTEX 2973. *Metabolic Engineering* **48**, 163-174, doi:10.1016/j.ymben.2018.06.002 (2018).
- 14 Ryan Davis, J. M., Christopher Kinchin, Nicholas Grundl, Eric C.D. Tan. Process Design and Economics for the Production of Algal Biomass: Algal Biomass Production in Open Pond Systems and Processing Through Dewatering for Downstream Conversion. Report No. NREL/TP-5100-64772, (2016).

Reviewers' Comments:

Reviewer #1:

Remarks to the Author:

Authors performed an additional work of the out-door cultivation for the analysis of SAC, which is now informative for the readers to evaluate the ML-assisted SAC system.

Reviewer #2:

Remarks to the Author:

The authors answered to all my comments and modified the manuscript accordingly. Therefore, I recommend the publication of this paper.

Reviewer #3:

Remarks to the Author:

The author added many details into the revised version and addressed all my comments. The revised version has great improvements and is recommended for publication in Nature Communications.